# Visualizing chaperonin function in situ by cryo-electron tomography

Jonathan Wagner[1,2,3,4], Alonso I. Carvajal[1], Andreas Bracher[1], Florian Beck[5], William Wan[6], Stefan Bohn[5,7], Roman Körner[1], Wolfgang Baumeister[2 ✉], Ruben Fernandez-Busnadiego[3,4,8 ✉] & F. Ulrich Hartl[1 ✉]

Chaperonins are large barrel-shaped complexes that mediate ATP-dependent protein folding[1–3]. The bacterial chaperonin GroEL forms juxtaposed rings that bind unfolded protein and the lid-shaped cofactor GroES at their apertures. In vitro analyses of the chaperonin reaction have shown that substrate protein folds, unimpaired by aggregation, while transiently encapsulated in the GroEL central cavity by GroES[4–6]. To determine the functional stoichiometry of GroEL, GroES and client protein in situ, here we visualized chaperonin complexes in their natural cellular environment using cryo-electron tomography. We find that, under various growth conditions, around 55–70% of GroEL binds GroES asymmetrically on one ring, with the remainder populating symmetrical complexes. Bound substrate protein is detected on the free ring of the asymmetrical complex, defining the substrate acceptor state. In situ analysis of GroEL–GroES chambers, validated by high-resolution structures obtained in vitro, showed the presence of encapsulated substrate protein in a folded state before release into the cytosol. Based on a comprehensive quantification and conformational analysis of chaperonin complexes, we propose a GroEL–GroES reaction cycle that consists of linked asymmetrical and symmetrical subreactions mediating protein folding. Our findings illuminate the native conformational and functional chaperonin cycle directly within cells.

The bacterial chaperonin GroEL cooperates with its cofactor GroES in assisting the folding of roughly 10% of newly synthesized proteins, including proteins with α/β topology that fail to fold spontaneously[1,2,7,8]. GroEL is a cylindrical complex of around 800 kDa containing two heptameric rings of 57 kDa subunits stacked back to back. The subunits consist of apical, intermediate and equatorial domains and a flexible C-terminal tail protruding into the ring cavity[9] (Fig. 1a, top left inset). The apical domains mediate substrate protein (SP) binding and the equatorial domains mediate ATP binding and hydrolysis. Hydrophobic residues at the apical domains recruit unfolded SP. ATP-dependent binding of the lid-shaped GroES (a heptamer of 10 kDa subunits), capping the SP-containing ring (the *cis*-ring), results in the burial of hydrophobic surfaces on GroEL and displaces the bound protein into an enclosed chamber. SP folds inside this chamber during ATP hydrolysis on the GroEL *cis*-ring, and a second SP can bind to the *trans*-ring. The *cis*-chamber opens following ATP binding to the *trans*-ring, dissociating GroES through negative inter-ring allostery to allow SP release[1,2,10–12]. Thus, the two rings of GroEL are sequentially folding active. However, in vitro studies[1,2,13] showed that GroES not only binds asymmetrically with GroEL ('bullet' complexes, EL–ES$_1$), but can also associate symmetrically with both rings ('football' complexes, EL–ES$_2$). Some reports have suggested that SP binding

shifts GroEL entirely from an asymmetrical cycle to a symmetrical mode[14].

The cell cytosol is characterized by a high degree of macromolecular crowding, which profoundly affects protein–protein interactions[15]. To investigate how the available in vitro data apply to the situation in the intact cell, here we explored the chaperonin mechanism within its natural cellular context by cryo-electron tomography (cryo-ET)—a technique enabling in situ visualization of macromolecular assemblies at subnanometre resolution[16–23]. We find that the native chaperonin cycle consists of linked asymmetrical and symmetrical subreactions mediating protein folding.

## GroEL–GroES complexes in situ

For visualization of GroEL by cryo-ET in situ, *Escherichia coli* BL21(DE3) cells were vitrified on electron microscopy grids and thinned by cryogenic focused ion beam milling before imaging (Fig. 1a,b and Extended Data Fig. 1a). EL–ES$_1$ and EL–ES$_2$ complexes were readily observed in raw tomograms (Fig. 1a, right insets), whereas GroEL alone was undetectable. We used template matching with reference structures for systematic identification and classification (Extended Data Fig. 1b), showing the relative proportions and cellular distribution of these

[1]Department of Cellular Biochemistry, Max Planck Institute of Biochemistry, Martinsried, Germany. [2]Research Group Molecular Structural Biology, Max Planck Institute of Biochemistry, Martinsried, Germany. [3]Institute of Neuropathology, University Medical Center Göttingen, Göttingen, Germany. [4]Cluster of Excellence "Multiscale Bioimaging: from Molecular Machines to Networks of Excitable Cells" (MBExC), University of Göttingen, Göttingen, Germany. [5]Research Group CryoEM Technology, Max Planck Institute of Biochemistry, Martinsried, Germany. [6]Vanderbilt University Center for Structural Biology, Nashville, TN, USA. [7]Institute of Structural Biology, Helmholtz Center Munich, Oberschleissheim, Germany. [8]Faculty of Physics, University of Göttingen, Göttingen, Germany. ✉e-mail: baumeist@biochem.mpg.de; ruben.fernandezbusnadiego@med.uni-goettingen.de; uhartl@biochem.mpg.de

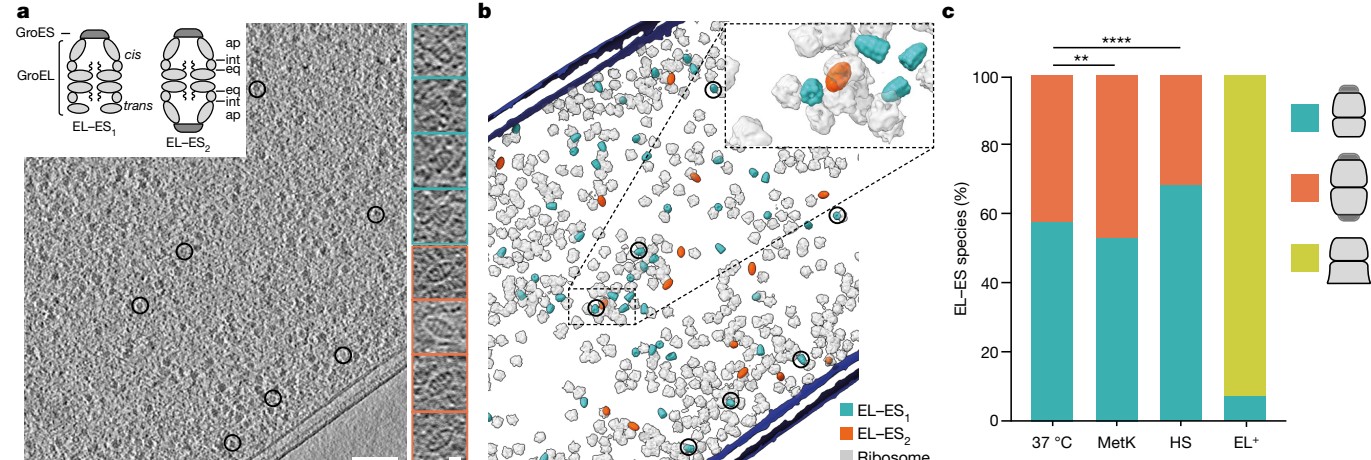

**Fig. 1 | In situ visualization and quantification of GroEL–GroES complexes.**
**a**, Left, z-slice of a representative tomogram of an *E. coli* cell exposed to HS
($n = 58$ tomograms). GroEL–GroES complexes are represented by black circles.
Top left inset, schematic representation of GroEL in asymmetrical (EL–ES₁)
and symmetrical (EL–ES₂) complexes with apical (ap), intermediate (int) and
equatorial (eq) GroEL domains indicated. The half of the EL–ES₁ complex bound
to ES is marked as *cis* and the opposing side as *trans*. Flexible C-terminal
sequences protruding into the GroEL cavity are indicated by wavy lines. Right,
gallery showing central subtomogram slices of EL–ES₁ and EL–ES₂ complexes in
side view. **b**, Three-dimensional rendering of EL–ES₁ complexes (blue), EL–ES₂

complexes (orange) and ribosomes (light grey) from the tomogram shown in **a**.
Cell membranes are depicted in dark blue. Complexes highlighted in **a** are
marked by black circles. **c**, Relative abundance of EL–ES₁ (blue), EL–ES₂ (orange)
and EL (yellow) complexes in tomograms from cells grown under differing
conditions, and also following MetK overexpression at 37 °C (MetK).
Differences in relative abundance are statistically significant, with *P* values
(Wilcoxon rank-sum test, two-sided) of 0.007 for MetK relative to 37 °C and
$5 \times 10^{-7}$ for HS relative to 37 °C. *P* values were not corrected for multiple testing
(37 °C, $n = 48$; HS, $n = 58$; MetK, $n = 60$ tomograms). Scale bars, 100 nm (**a**),
10 nm (**a**, right inset).

complexes. In cells growing at 37 °C, EL–ES₁ and EL–ES₂ complexes
occurred at an approximate ratio of 60:40% (Fig. 1c). To validate the
accuracy of the template-matching results we compared the numbers
of identified chaperonin complexes with those of ribosomes, which can
be readily identified in cryo-ET[23,24]. We localized essentially all cellular
ribosomes (Extended Data Fig. 2a,b), and determined a median ratio
of GroEL to ribosomes of 1:23 during growth at 37 °C (Extended Data
Fig. 2c). Quantification by mass spectrometry (MS) confirmed these
results (Extended Data Fig. 2c, blue crosses), indicating that our cryo-ET
analysis had identified most GroEL complexes. However, owing to the
inherent limitations of template matching, we cannot rule out a small
fraction of false-positive or false-negative particles.

To load GroEL with chaperonin-dependent SP, we first increased
the level of both GroEL and GroES by around sixfold (Extended Data
Fig. 3a–c), to reduce occupancy with endogenous SP, and then strongly
overexpressed the obligate GroEL substrate *S*-adenosylmethionine
synthase (MetK)[25,26] (Extended Data Fig. 3d,e). Biochemical analysis by
GroEL immunoprecipitation and MS demonstrated that, on average,
about 1.3 MetK molecules bound per GroEL complex, corresponding
to over 50% of GroEL rings containing MetK (Extended Data Fig. 3f,g).
The relative abundance of EL–ES₁ and EL–ES₂ complexes in tomograms
was about 55% and 45%, respectively, similar to growth without MetK
overexpression (Fig. 1c).

To explore changes in chaperonin function under stress, we exposed
cells to heat stress (HS) at 46 °C for 2 h. Note that *E. coli* grows efficiently
under HS in full medium (Extended Data Fig. 3d) although numerous
proteins are destabilized[27], increasing the demand for chaperonin. HS
induced a roughly threefold increase in GroEL and GroES abundance
(Extended Data Fig. 3a,b), with a ratio of GroEL to ribosomes of about
1:10 in MS and cryo-ET data (Extended Data Fig. 2c). Notably, the level
of EL–ES₁ complexes increased to 70% of total (Fig. 1c); GroEL alone
remained undetectable. Thus, HS promotes the formation of asym-
metric chaperonin complexes.

We next investigated whether EL–ES₂ complexes form as a conse-
quence of GroES:GroEL concentration ratio. Expression of the *groES*
and *groEL* genes (*groESL*), organized in an operon[28], resulted in an
approximate 1:1 GroES:GroEL ratio[29], equivalent to around a twofold

excess of GroES (7-mer) over GroEL (14-mer)[30], with both proteins being
essential[28]. To reverse the physiological ratio of GroES and GroEL we
selectively overexpressed GroEL (EL⁺ cells) at 37 °C, resulting in a
roughly 4.5-fold increase in GroEL (Extended Data Fig. 3h). EL⁺ cells
grew essentially as wild-type (WT) (Extended Data Fig. 3d) but con-
tained only free GroEL (EL complex) and EL–ES₁ (around 90% and 10%
of total GroEL, respectively) and no EL–ES₂ complexes (Fig. 1c). Nota-
bly, because EL–ES₁ complexes were of similar abundance relative to
ribosomes as in WT cells (Extended Data Fig. 3i), the absence of EL–ES₂
resulted in a reduction in the overall level of GroEL–GroES complexes.
Nevertheless, overexpression of MetK did not impair the growth of EL⁺
cells (Extended Data Fig. 3e).

In summary, asymmetrical and symmetrical chaperonin complexes
coexist in vivo, with EL–ES₁ predominating under all growth conditions
tested, including high SP load and HS. Cells grew efficiently when EL–ES₂
complexes were not populated, indicating that EL–ES₁ complexes are
sufficient for function.

## In situ structures of chaperonin complexes

Subtomogram averaging (STA) produced structural models for EL–
ES₁, EL–ES₂ and EL complexes at around 10–12 Å resolution follow-
ing the application of symmetry (Fig. 2a–d, Extended Data Fig. 4a–d
and Extended Data Table 1). Molecular models were derived, starting
from rigid-body fitting of high-resolution GroEL structures. EL–ES₁
complexes were further classified based on the positioning of the api-
cal domains of the GroEL *trans*-ring, resulting in two conformations
referred to as 'narrow' and 'wide' (Fig. 2a,b,e,f). In the narrow state
the opening of the *trans*-ring has a diameter of around 45 Å (Fig. 2f),
similar to the EL–ADP₇–ES₁ crystal structure (PDB 1AON[31]) (Extended
Data Fig. 4e). By contrast, the wide conformation shows a significant
reorientation of the apical domains, extending the ring opening to
around 65 Å (Fig. 2f), which would facilitate the exit of larger SPs such
as folded MetK (approximately $70 \times 60 \times 30$ Å³ in size). Consistent
with this interpretation, a similar conformation was observed in a
cryo-electron microscopy (cryo-EM) structure of EL–ES₁ with bound
ADP (PDB 7PBJ[32]) (Extended Data Fig. 4f). Under all conditions analysed,

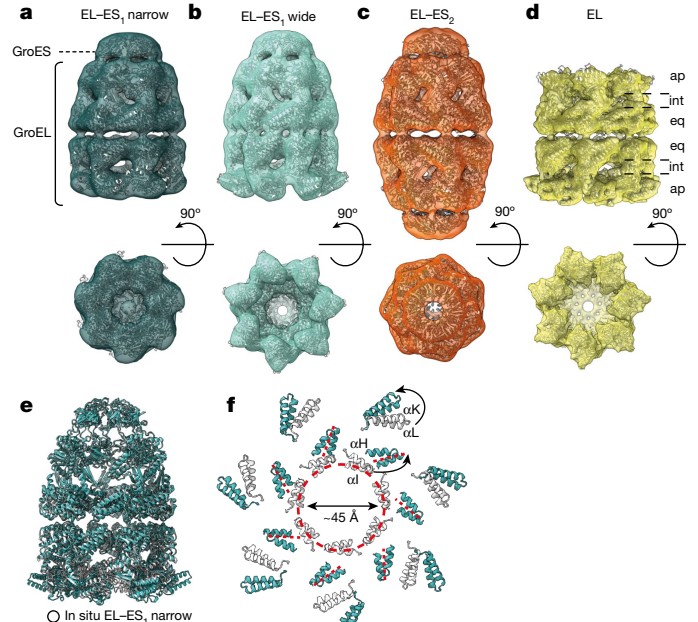

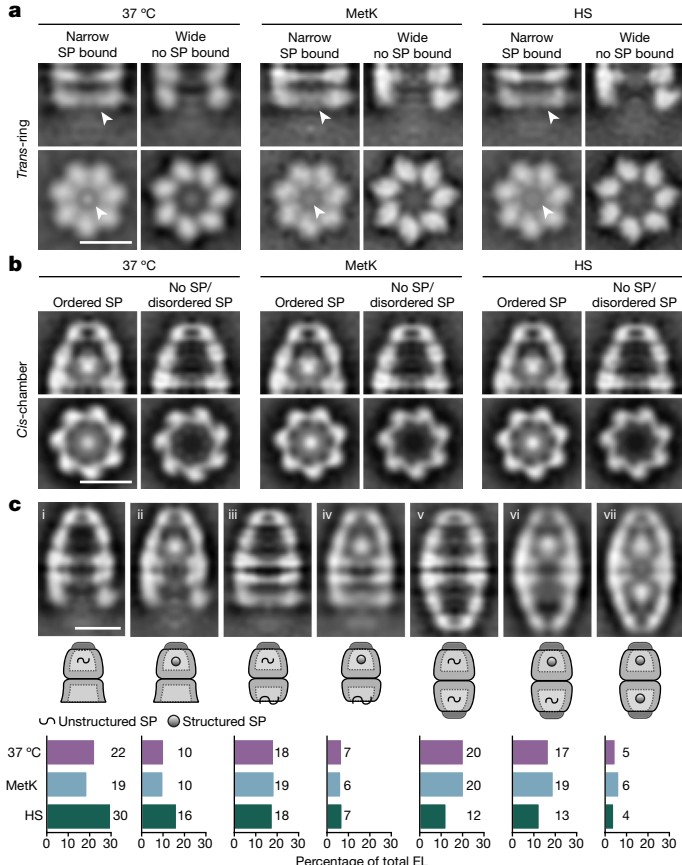

**Fig. 2 | In situ structures of chaperonin complexes. a–d**, Subtomogram averages of EL–ES$_1$ narrow (dark blue) (**a**), EL–ES$_1$ wide (light blue) (**b**), EL–ES$_2$ (orange) (**c**) and EL (yellow) (**d**) complexes (symmetry applied) at nominal resolutions of 10–12 Å (Extended Data Fig. 4a–d). Side and top views are shown. Ribbon representations of the models derived from STA densities for narrow EL–ES$_1$, wide EL–ES$_1$, EL–ES$_2$ and open EL, respectively, are superposed. EL–ES$_1$ and EL–ES$_2$ complexes are derived from tomograms of cells grown at 37 °C, exposed to HS or following overexpression of MetK; EL complexes are derived from tomograms of cells with GroEL overexpression (EL$^+$). The locations of GroES and GroEL rings are indicated in **a**; positions of the apical, intermediate and equatorial domains in GroEL rings, respectively, are indicated in **d**. **e**, Overlay of wide EL–ES$_1$ in situ structure model (light blue) and the narrow EL–ES$_1$ in situ structure model (white) by least-squares fitting of equatorial domains. Structures are shown in ribbon representation. **f**, Widening of the *trans*-ring opening from around 45 to 65 Å between in situ EL–ES$_1$ complexes with narrow and wide *trans*-ring. Only the SP-binding helices αI and αH and helical hairpins αL and αK are shown. Red dashed lines indicate the SP-binding groove; curved black arrows denote reorientation of the respective domains.

the wide *trans*-ring conformation was more abundant than the narrow state, especially following HS (Extended Data Fig. 4g).

The in situ structure of EL–ES$_2$ at the given resolution showed no major deviations from the crystal structure of the non-cycling symmetrical complex with bound ADP–BeF$_x$ (PDB 4PKO[33]) (Extended Data Fig. 4h). Interestingly, in the in situ structure of GroEL alone at a resolution of about 9.8 Å (Extended Data Fig. 4d)—attained following GroEL overexpression (EL$^+$)—one ring mirrored the wide *trans*-ring conformation of EL–ES$_1$ whereas the other was in a more narrow state (Fig. 2d) with continuous additional density at the apical domains (Extended Data Fig. 4i,j). This density probably resulted from symmetry-averaged, unfolded SP that had accumulated on GroEL at substoichiometric GroES. Thus the GroEL complex shows intrinsic inter-ring asymmetry in vivo, reflecting the negative allosteric coupling between rings and leading to preferential substrate binding to one ring.

## Visualization of substrate in the GroEL–GroES cycle

Similar to GroEL alone, the *trans*-ring of EL–ES$_1$ in the narrow state also contained central density at the apical domains (indicated by arrowheads in Fig. 3a), presumably representing bound SP before encapsulation by GroES. No SP density was observed in the wide *trans*-ring, nor was a narrow state without bound SP resolved (Fig. 3a). Indeed, the

**Fig. 3 | Densities of substrate proteins in in situ structures. a**, Slices through STA densities of the *trans*-ring of EL–ES$_1$ complexes from 37 °C, MetK and HS cells in side view (top) and top view (bottom). For all growth conditions, classification resulted in two distinct classes of EL–ES$_1$ *trans*-rings: one with a narrow *trans*-ring containing a strong, localized density at the level of the apical GroEL domains (left), and one with a wide *trans*-ring and no extra density (right). **b**, Vertical and horizontal slices through STA densities of GroEL–GroES chambers from 37 °C, MetK and HS cells at the level of SP density. Processing resulted in two distinct classes of GroEL–GroES chamber: one containing a strong, localized density near the bottom of the chamber and one with only a weak, delocalized density in the chamber. Following splitting of particles based on growth conditions (37 °C, HS, MetK), the same two classes were found in all three groups. Subsequent experiments led to the assignment of encapsulated SP as either ordered or disordered. **c**, Vertical slices through the centre of STA densities of all GroEL–GroES species found in situ with different conformational states and SP occupancy (top), together with their relative abundance (bottom). Species i and ii are EL–ES$_1$ complexes with a *trans*-ring in the wide conformation and a *cis*-ring with either disordered or no SP (i) or folded SP (ii). Species iii and iv are EL–ES$_1$ complexes with a *trans*-ring in narrow conformation and *cis*-rings with either disordered or no SP (iii) or folded SP (iv). Species v–vii are EL–ES$_2$ complexes with either no or disordered SP (v), folded SP in one chamber (vi) or folded SP in both chambers (vii), as shown schematically in pictograms. Scale bars, 10 nm. Schematic in panel **c** adapted from ref. 1, Elsevier.

apical domains in the narrow state expose the functionally critical hydrophobic residues in helices αI and αH, forming a continuous furrow for SP binding[34], whereas in the wide state the coherent binding surface was disrupted (Fig. 2f). Thus, following GroES dissociation, the *trans*-ring in its wide conformation would allow SP release whereas binding of new SP presumably occurs following conversion to the narrow conformation. Interestingly, the ratio of EL–ES$_1$ with wide *trans*-ring to EL–ES$_1$ with narrow *trans*-ring (Extended Data Fig. 4g) correlated closely with the overall ratio of EL–ES$_1$ to EL–ES$_2$ (Fig. 1c). This suggests that binding of SP to the *trans*-ring may facilitate the formation of symmetrical complexes by lowering negative inter-ring allostery[14].

Furthermore, because EL–ES₁ species with SP bound in *trans* are populated, association of the second GroES must be a relatively slow step.

Next, for visualization of encapsulated SP we extracted and pooled GroEL–GroES chambers from all EL–ES₁ and EL–ES₂ complexes and analysed them by averaging and three-dimensional classification of the chamber interior (Extended Data Fig. 4k). For each growth condition we identified two distinct classes of complex (Fig. 3b): the GroEL–GroES chambers of class I contained a well-defined globular density close to the bottom of the cavity, consistent with structured SP. The chambers of class II showed only a weak and fuzzy density, representing empty cavities and/or the presence of dynamic, non-native SP conformations that would be obscured by averaging.

Sorting the EL–ES₁ and EL–ES₂ complexes in the in situ datasets according to the presence of encapsulated and/or bound SP allowed us to quantify a total of seven different states of EL–ES₁ and EL–ES₂ (Fig. 3c). At 37 °C growth, the relative proportions of these species were largely independent of MetK overexpression, with a subset of EL–ES₂ complexes containing structured SP in both chambers. Interestingly, following HS, EL–ES₁ complexes with wide *trans*-ring conformation (no bound SP) were enriched (Fig. 3c(i,ii)) and EL–ES₂ complexes reduced (Fig. 3c(v–vii)), perhaps due to changes in the ATP:ADP ratio during HS[35]. This is consistent with SP binding to the *trans*-ring facilitating EL–ES₂ formation.

These results define the chaperonin species that are populated in vivo and demonstrate that complexes EL–ES₁ and EL–ES₂ are both functionally active.

## Structure of MetK inside chaperonin

To what extent does SP fold inside the chaperonin chamber during the functional GroEL–GroES cycle in vivo? Previous in vitro cryo-EM analyses of encapsulated client protein under non-cycling conditions had shown a distinct density in the equatorial half of the chamber, representing SP folding intermediates at low resolution[36–39]. We performed a similar in vitro analysis on encapsulated MetK, by both cryo-EM and cryo-ET, for comparison with the in situ cryo-ET structures. We prepared SP-bound GroEL by heat denaturation of MetK in the presence of GroEL[40]. Encapsulation occurred following the addition of GroES and ATP-BeF$_x$ (Extended Data Fig. 5a,b). BeF$_x$ favours the formation of stable (non-cycling) EL–ES₂ complexes with bound ADP–BeF$_x$ (ref. 41). MS analysis indicated a stoichiometry of MetK to GroEL 14-mer of roughly 1.2 (Extended Data Fig. 5c), similar to MetK overexpression (Extended Data Fig. 3f,g). Reference-free, two-dimensional classification demonstrated the presence of EL–ES₂ as well as some EL–ES₁ complexes (Extended Data Figs. 5d and 6a–d). The latter exhibited subpopulations with wide and narrow *trans*-ring conformations resembling those observed in situ (Extended Data Fig. 6e,f), with density for bound SP in the narrow state (Extended Data Fig. 6f).

For visualization of encapsulated SP, GroEL–GroES chambers were processed for cryo-EM structure determination (Extended Data Figs. 5d and 6c,d). Alignment and classification showed that around 40% of GroEL–GroES units contained density for an ordered MetK molecule close to the equatorial region of the chamber (Fig. 4a–d, Extended Data Fig. 7 and Extended Data Table 1). The remainder contained only a faint, smeared-out density, representing empty chambers and chambers with incompletely folded or misaligned MetK. The substructure of the ordered MetK molecules was solved at a resolution of approximately 3.7 Å, showing side-chain density in its hydrophobic core (Extended Data Fig. 7d–f). The encapsulated MetK was native-like, with a root mean squared deviation relative to the crystal structure (PDB 7LOO[42]) of 1.4 Å for 366 of the 379 Cα atoms (Fig. 4e). The main difference was in the conformation of residues 97–111, the so-called core loop. This region packs against bound *S*-adenosylmethionine and an adjacent subunit in the MetK tetramer[42,43] (Extended Data Fig. 8a), but in the encapsulated MetK subunit adopted a more extended conformation

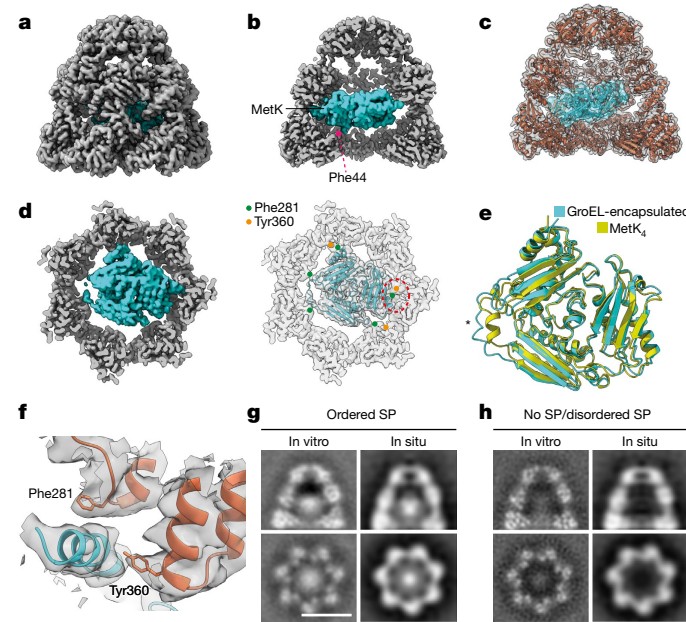

**Fig. 4 | Structure of MetK inside the GroEL–GroES chamber. a–c**, Structure of GroEL–GroES chambers with folded, encapsulated MetK (teal). Side view of the density (**a**), side view of the chamber interior (**b**) and superposition of the molecular model in ribbon representation (**c**) are shown. One of the contacts between MetK and a Phe44 residue of GroEL is indicated in **b**. **d**, Cut-away representations showing a top view of the density map (left) and the superposed MetK model (teal) in ribbon representation (right). GroEL contact residues Phe281 and Tyr360 are indicated by green and orange dots, respectively (right). A red dotted ellipse marks the area magnified in **f**. Phe44 residues are not visible in the slice shown. **e**, Overlay of the structures of GroEL–GroES-encapsulated MetK (teal) and a subunit from the isolated MetK tetramer (PDB 7LOO[42], yellow). Asterisk marks the core loop of MetK. **f**, Detailed view of a contact between MetK and GroEL in the region marked in **d** (right). Contact residues Phe281 and Tyr360 are shown as sticks. **g**,**h**, Slices through SPA maps obtained in situ and in vitro of GroEL–GroES chambers with ordered SP (**g**) and no/disordered SP (**h**). Grey values were normalized to the GroEL–GroES chamber for all panels. Note that these maps were symmetry averaged. Scale bar, 10 nm.

that was not well resolved (Fig. 4e). The core loop apparently remains unstructured until tetramer assembly following release from chaperonin. The encapsulated MetK makes multiple contacts with the GroEL cavity wall, contacting two subunits at Phe44 in the equatorial GroEL domain as well as five subunits at Phe281 and three at Tyr360, both protruding from the apical GroEL domains (Fig. 4b–d). These residues appear to interact with MetK via van der Waals contacts. However, the side chains of the interacting residues are poorly defined, indicating heterogeneity in these regions of the structure (Fig. 4f). The GroEL subunits contacting MetK show only minor conformational rearrangements, with root mean squared deviation values of 0.5–1.0 Å compared with a new 2.5 Å cryo-EM structure of empty GroEL–(ADP–BeF$_x$)₇–ES chambers (Extended Data Fig. 8b–g and Extended Data Table 1). Of note, the GroEL cavity wall does not contact the interface regions of the MetK subunit that become buried following assembly. These regions apparently remain solvent exposed in the chamber (Extended Data Fig. 8a) but could be reached by flexible C-terminal Gly–Gly–Met repeat sequences (23 residues) of the GroEL subunits not resolved in the cryo-EM structure.

To further rationalize our in situ cryo-ET analysis of encapsulated SP (Fig. 3b), we next performed cryo-ET on isolated GroEL–GroES–MetK complexes using the same imaging parameters as for in situ tomography (Extended Data Table 1). In agreement with the single-particle data, the classification of chambers within these complexes again yielded

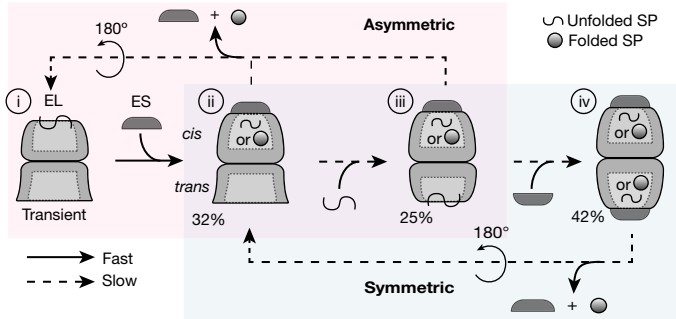

**Fig. 5 | Mechanism for GroEL–GroES-assisted protein folding in vivo.**
The interconnected reaction cycles involving asymmetric and symmetric chaperonin complexes are highlighted by a light red and light blue background, respectively. Folded SP is depicted as a sphere and unfolded SP as a wriggle. Steps assumed to be fast or slow are indicated by arrows with solid and dashed lines, respectively. Numbers below the pictograms indicate the fraction of total of the respective complex at 37 °C. The abundance of GroEL alone was estimated to be below 5%. Adapted from ref. 1, Elsevier.

two classes. Class I (around 40% of particles) contained a strong density in the chaperonin cavity, corresponding well with symmetry-averaged folded MetK (Fig. 4g, left), whereas class II chambers (roughly 60% of particles) showed a weak, diffuse density (Fig. 4h, left). The location of the structured MetK near the equatorial region of the GroEL–GroES chamber and its density relative to the GroEL wall (Fig. 4g, left) coincided with that of the folded SP in situ (Fig. 4g, right). Specifically, the position of the SP centre of mass following MetK overexpression, in which MetK is highly enriched on GroEL (Extended Data Fig. 3f,g), was in almost perfect agreement with the position of the folded MetK in the in vitro tomograms (Extended Data Fig. 9). Although other SPs besides MetK may be present within GroEL in situ, our data suggest that these proteins occupy a similar location within the chamber. Thus, encapsulation in vivo resulted in SP folding to a native or native-like, compact state.

## Conclusions

Our analysis of GroEL–GroES complexes in situ using cryo-ET allows us to define the intermediate steps of the bacterial chaperonin cycle in vivo. We find that both asymmetric and symmetric chaperonin complexes operate in linked subreactions (Fig. 5). GroEL without bound GroES is below detectability and may exist only transiently (Fig. 5(i)). By contrast, asymmetric EL–ES$_1$ with and without bound SP on the *trans*-ring is abundant, defining the main SP acceptor state (Fig. 5(ii)). In the asymmetric reaction the GroEL rings alternate between folding active and binding active. Following GroES dissociation, SP exits the folding chamber (Fig. 5(iii–i)), facilitated by a wide conformation of the apical GroEL domains, possibly generating a short-lived GroEL-only intermediate (Fig. 5(i)). Alternatively, rather than completing the asymmetric cycle, GroES binding to the *trans*-ring gives rise to EL–ES$_2$ (Fig. 5(iii–iv)), in which both rings can be folding active. Because folding begins in the *cis*-chamber of EL–ES$_1$ and can continue in the EL–ES$_2$ complex, the symmetric cycle may benefit SPs with slow folding kinetics[13].

How is the partitioning between asymmetric and symmetric chaperonin reactions regulated? In the canonical asymmetric cycle in vitro the GroEL rings are coupled by negative allostery, with ATP binding to the *trans*-ring causing ADP and GroES release from the *cis*-ring (Fig. 5 (species ii/iii–i))[1,44,45]. Negative inter-ring allostery also operates in vivo, favouring EL–ES$_1$ formation, because exclusively EL–ES$_1$ complexes mediate protein folding at GroEL excess over GroES. In WT cells, EL–ES$_2$ complexes are also functional. Conversion of the EL–ES$_1$ *trans*-ring

from a wide conformation to the narrow, SP-binding state (Fig. 5(ii–iii)) appears to be limiting for EL–ES$_2$ formation (Fig. 5(iii–iv)), because the ratio of EL–ES$_1$ wide to EL–ES$_1$ narrow correlates closely with the overall EL–ES$_1$ to EL–ES$_2$ ratio.

Our cryo-ET analysis also demonstrated that, before release into bulk cytosol, SP reaches a folded state in the GroEL–GroES chamber. To validate this finding we solved as a reference the structure of stably encapsulated MetK, an obligate GroEL substrate[26], in vitro. The MetK subunit is natively folded and is located close to the equatorial region of the GroEL–GroES cavity[36,38,39]. Encapsulated MetK makes weak contacts with specific GroEL residues (Fig. 4) and is in close proximity to flexible C-terminal GGM repeat sequences of the equatorial GroEL domains, which may promote efficient folding[46,47]. The position and density of folded, encapsulated MetK closely resemble those of structured SP in the GroEL–GroES chamber in situ.

In summary, our analysis provides a detailed view of the chaperonin reaction cycle in vivo, in which asymmetric and symmetric GroEL–GroES complexes are functionally linked. SP accumulates inside the chaperonin chamber in a folded state before release into cytosol.

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

# Methods

## Plasmids and strains

*Escherichia coli* BL21(DE3) Gold cells (Stratagene) were used for growth analysis, electron tomography and protein expression. For tomography and biochemical experiments, GroEL was expressed from a pBAD33 plasmid containing the *groEL* gene under the control of an araBAD promotor (EL[+] cells)[26]. For overexpression of GroEL and GroES, a pBAD33 plasmid containing both *groEL* and *groES* genes under the control of an araBAD promotor was used[48]. MetK was expressed from a pET22b plasmid previously described[26].

## Antibodies

Polyclonal antisera used against GroEL, GroES, MetK and GAPDH were previously described[26], and the rabbit antiserum against α-lactalbumin was a product of East Acres Biologicals immunization service.

## *E. coli* growth

*E. coli* cells were grown in lysogeny broth (LB) medium that contained, depending on the plasmids used, the antibiotics ampicillin (200 μg ml[−1], pET22b-MetK) and chloramphenicol (32 μg ml[−1], pBAD33 variants). For overexpression of GroEL, GroES and MetK (MetK cells), transformed *E. coli* Bl21 (DE3) pBAD33-GroEL:ES pET22b-MetK cells were grown to early exponential phase at 37 °C, and GroEL–GroES expression using the pBAD33 promoter was induced for 90 min by supplementation of LB medium with arabinose to a final concentration of 0.2% (w/v). Cells were subsequently harvested by centrifugation at 8,000*g* (4 °C for 10 min) and resuspended to an optical density (600 nm, $OD_{600}$) of 0.1–0.2 in fresh LB medium containing both antibiotics and 1 mM iso-propyl β-D-thiogalactopyranoside (IPTG), to induce MetK expression under control of the T7 promoter for 40 min. GroEL expression (EL[+]) was induced in *E. coli* Bl21 (DE3) pBAD33-GroEL by supplementation of LB medium with arabinose to a final concentration of 0.1% (w/v) and growth of the culture at 37 °C. To expose *E. coli* Bl21 (DE3) cells to HS, cells were first cultured to early exponential phase at 37 °C and then incubated in a shaking water bath at 46 °C for 2 h.

## *E. coli* growth curves

Cells were cultured as described above. Aliquots were removed at the time points indicated for optical density measurement at $OD_{600}$. To ensure exponential growth conditions, growing cultures were diluted to an $OD_{600}$ of 0.1 with prewarmed LB medium containing the necessary antibiotics and arabinose when $OD_{600}$ just exceeded 0.4. Growth curves for MetK and EL[+]/MetK cells were measured following termination of GroEL induction by transfer of cells into arabinose-free medium containing 1 mM IPTG for MetK overexpression. The first sample was taken 5 min after changing the medium. Data were processed for fitting in R.

## Protein expression and purification

GroEL, GroES and MetK proteins were expressed and purified as previously described[26,49].

## Measurement of protein concentration

Concentrations of purified proteins were determined by measurement of absorbance at 280 nm using absorbance coefficients calculated from the protein sequence with the program ProtParam[50]. Protein concentrations of cell lysates were determined with the Pierce Coomassie Plus (Bradford) Assay Kit (Thermo Fisher Scientific) as described by the manufacturer.

## Preparation of cell lysates

Cultures were prepared as described above, harvested by centrifugation and the cell pellet flash-frozen in liquid nitrogen before further processing. Spheroplasts were prepared at 4 °C as previously described[51]. In brief, cells were resuspended in 100 mM Tris-HCl pH 8.0 and washed twice with 2 ml of buffer. The pellet was then resuspended in HMK buffer (50 mM HEPES-KOH pH 7.2, 20 mM Mg acetate, 50 mM K acetate) supplemented with 20% (w/v) sucrose and 0.25 mg ml[−1] lysozyme. Cells were then incubated on ice for 7 min and transferred to 37 °C for 10 min. The resulting suspension was supplemented with Complete EDTA-free protease inhibitor cocktail (Roche), and spheroplasts were lysed by the addition of 0.1% (v/v) Triton X-100 and subsequent sonication.

## Mass spectrometry

Cell lysates were reduced by the addition of dithiothreitol (DTT) to a final concentration of 10 mM and heated to 56 °C for 45 min. Acylation of thiol groups was performed by the addition of chloroacetamide to a final concentration of 55 mM and incubation for 45 min in the dark, followed by a first digestion step with Lys-C (Wako) at a w/w ratio of 1:20 for 2 h at 37 °C. This was followed by a second digestion step overnight with trypsin (Roche) at a 1:20 (w/w) ratio at 37 °C. The reaction was stopped by the addition of trifluoroacetic acid to a final volume of 1%. Peptides were desalted using OMIX C18 (100 μl) tips (Agilent Technologies, no. A57003100) according to the manufacturer's instructions.

Desalted peptides were dissolved in 12 μl of 5% formic acid, sonicated in an ultrasonic bath, centrifuged and transferred to autosampler vials (Waters). Samples were analysed on an Easy nLC-1200 nanoHPLC system (Thermo) coupled to a Q-Exactive Orbitrap HF mass spectrometer (Thermo). Peptides were separated on pulled-spray columns (ID 75 μm, length 30 cm, tip opening 8 μm, NewObjective) packed with 1.9 μm C18 particles (Reprosil-Pur C18-AQ, Dr Maisch) using either a stepwise 196 min gradient (comparison of 37 °C, HS and MetK) or a stepwise 67 min gradient (all other samples) between buffer A (0.2% formic acid in water) and buffer B (0.2% formic acid in 80% acetonitrile). Samples were loaded on the column by the nanoHPLC autosampler at a pressure of 900 bar. The high-performance liquid chromatography flow rate was set to 0.25 μl min[−1] during analysis. No trap column was used. The following parameters were used for comparison of growth conditions 37 °C, HS and MetK: MS, resolution 60,000 (full-width at half-maximum (FWHM) setting); MS mass range 300–1,650 *m/z*; MS-AGC-setting $3 × 10^6$; MS-MaxIT 50 ms; MS/MS fragmentation of the 15 most intense ions (charge state 2 or higher) from the MS scan; MS/MS resolution 15,000 (FWHM setting); MS/MS-AGC-setting $10^5$; MS/MS-MaxIT 50 ms; MS/MS isolation width 1.8 *m/z*; collision-energy setting 29 (NCE). All other samples were analysed with the following parameters: MS resolution 120,000 (FWHM setting); MS mass range 300–1,650 *m/z*; MS-AGC-setting $3 × 10^6$; MS-MaxIT 100 ms; MS/MS fragmentation of the ten most intense ions (charge state 2 or higher) from the MS scan; MS/MS resolution 15,000 (FWHM setting); MS/MS-AGC-setting $10^5$; MS/MS-MaxIT 50 ms; MS/MS isolation width 1.2 *m/z*; collision-energy setting 29 (NCE).

**MS data analysis.** Protein identification was performed using MaxQuant with default settings. The *E. coli* K12 strain sequences of UNIPROT (v.2023-03-01) were used as the database for protein identification (Supplementary Information). MaxQuant uses a decoy version of the specified UNIPROT database to adjust false discovery rates for proteins and peptides below 1%.

## Quantification of MetK binding to GroEL

To quantify the fraction of GroEL with bound MetK in MetK-overexpressing cells, we immunoprecipitated GroEL with GroEL antibody followed by GroEL and MetK immunoblotting and liquid chromatography–tandem mass spectrometry. Cells were prepared and lysed as described above, but with the addition of apyrase (25 U ml[−1] final concentration) to rapidly deplete the ATP pool in the lysate and arrest the GroEL reaction cycle[26]. The lysate was clarified by centrifugation at 16,000*g* (4 °C for 10 min). Either 20 μl of a non-specific antibody (against α-lactalbumin) or a GroEL-specific antibody was coupled to 100 μl of recombinant protein A Sepharose 4B beads

(Thermo Fisher Scientific) as described by the manufacturer. The beads were loaded with sample (180 μg of protein) and incubated in 650 μl of HMK buffer for 1 h. The beads were washed twice with 600 μl of HMK buffer and then twice more with HMK containing 0.1% Triton X-100. For immunoblotting, elution was performed with 50 μl of 2× lithium dodecyl sulfate (Pierce) containing β-mercaptoethanol 5% (v/v) as prescribed by the manufacturer. For liquid chromatography–tandem mass spectrometry analysis, elution and digestion were performed with the IST MS sample preparation kit (Preomics) using the manufacturer's on-bead digestion protocol. Mass spectrometry was performed as described above.

## SDS–PAGE and immunoblotting

Before SDS–polyacrylamide gel electrophoresis (SDS–PAGE) analysis, cells were resuspended in HMK buffer supplemented with 2 mM DTT, 1 mM EDTA and 5% glycerol and subsequently sonicated, followed by centrifugation (20 min, 16,000$g$ at 4 °C). Protein samples were separated by electrophoresis on NuPAGE 10% Bis-Tris SDS gels (Invitrogen) using NuPAGE MES SDS running buffer (Invitrogen) at 150 V. Proteins were transferred to polyvinylidene difluoride membranes in blotting buffer (25 mM Tris, 192 mM glycine, 20% methanol) at 150 mA. Membranes were first incubated with primary antibodies in TBST buffer overnight at 4 °C and subsequently with horseradish peroxidase-conjugated secondary antibody for chemiluminescence detection. Uncropped immunoblots are provided in the Source Data file to Extended Data Fig. 3.

## In situ cryo-ET analysis

Cell cultures were grown as described above. For cryo-ET analysis, cells in exponential growth (approximate $OD_{600}$ 0.4) were rapidly (for about 2 min) concentrated to an approximate $OD_{600}$ of 10 by centrifugation at 8,000$g$ and subsequently applied to R 2/1 100 Holey carbon film Cu 200 mesh grids (Quantifoil) that were previously plasma cleaned for 30 s. The sample was blotted for 9 s at force 10 and then plunge-frozen in a mixture of liquid ethane and propane cooled by liquid nitrogen using a Vitrobot Mark IV (Thermo Fisher Scientific) at 70% humidity and 22 °C. Frozen grids were transferred to a dual-beam, cryo-focused ion beam (FIB)/ scanning electron microscope (Thermo Fisher Scientific; either Scios, Quanta, Aquilos or Aquilos 2). Cells were coated with a layer of inorganic platinum, if available in the system used, followed by the deposition of organometallic platinum using an in situ gas injection system (working distance, 10 mm; heating, 27 °C; time, 8 s). Removal of bulk material was done at a stage angle of 20–25° using gallium ions at 30 kV, 0.5 nA. Fine milling of lamellae was done at 11–13° stage tilt with successively lower currents between 0.3 nA and 30 pA, aiming for a final thickness of 100–200 nm (ref. 52). Lamellae for the selective GroEL overexpression dataset were prepared using Serial FIB[53], and an additional layer of inorganic platinum was added following fine milling to avoid charging during image acquisition[54]. The resulting lamellae were transferred to a TEM (Titan Krios, field emission gun 300 kV, Thermo Fisher Scientific) equipped with an energy filter (Quantum K2, Gatan), a direct detection camera (K2 Summit, Gatan), and tomograms were acquired at a magnification of ×42,000 (pixel size 3.52 Å), defocus ranging from −5.0 to −3.0 μm and the energy filter slit set to 20 eV using SerialEM 3.9.0 (ref. 55). Tomograms were recorded in dose-fractionated super-resolution mode, with a total dose of roughly 120 e⁻/Å² per tilt series. A dose-symmetric tilt scheme was used with an increment of 2–3° in a total range of ±60° from a starting angle of approximately 10° to compensate for lamellar pretilt (mostly around 11°)[56]. Frames were aligned using MotionCor2 (v.1.4.0, https://emcore.ucsf.edu/ucsf-software)[57]. The reconstruction was performed in IMOD using patch tracking (v.4.11.1, RRID:SCR_003297, https://bio3d.colorado.edu/imod/)[58] using the TOMOgram MANager (TOMOMAN) wrapper scripts[59]. Tilt-series images were dose filtered using TomoMAN's implementation of the Grant and Grigorieff exposure filter[60]. Defocus was estimated using CTFFIND4 (ref. 61).

Tomograms of the EL⁺ dataset were acquired on a Krios G4 equipped with a Selectris X energy filter and Falcon 4 direct electron detector (Thermo Fisher Scientific). Tilt series were collected with a dose-symmetric tilt scheme using TEM Tomography 5 software (Thermo Fisher Scientific). A tilt span of ±60° was used with 2° steps, starting at ±10°, to compensate for lamellar pretilt. Target focus was changed for each tilt series in steps of 0.5 μm over a range of −2.5 μm to +5 μm. Data were acquired in EER mode of Falcon 4 with a calibrated physical pixel size of 3.02 Å and a total dose of 3e⁻/Å² per tilt over ten frames. A 10 eV slit was used for the entire data collection. Data were preprocessed using TOMOMAN[59]. EER images were motion corrected using RELION's implementation of MotionCor2 (ref. 62). Defocus was estimated using CTFFIND4 (ref. 61). Reconstruction was performed with IMOD using local deposits of the inorganic platinum that was applied by sputtering following milling as fiducials. All tomograms were reconstructed using NovaCTF[63].

*E. coli* membranes were segmented for visualization using TomoSegMemTV 1.0.

## Cryo-ET analysis of in vitro reconstituted GroEL–GroES complexes

For generation of a GroEL–GroES reference for in situ tomographic analysis containing a defined substrate protein in a folded state and in a known topology, we imaged in vitro reconstituted GroEL–GroES–MetK complexes using the same data collection strategy and parameters as above for WT cells.

## Subtomogram averaging

For subtomogram averaging, all datasets acquired on the same microscope (37 °C, HS, MetK) were combined and processed together; the EL⁺ dataset was processed separately. The overall processing workflow is depicted in Extended Data Fig. 1b.

For template matching, PDB entry 1AON was used for EL–ES$_1$, 4PKO for EL–ES$_2$ and 5MDZ for 70S ribosomes to generate templates at a resolution of 40 Å using the molmap[64] command in Chimera[65]. Initial positions for a subset of EL–ES$_1$ and EL–ES$_2$ complexes and ribosomes were determined using the noise correlation template-matching approach implemented in STOPGAP, by fourfold binning to a pixel size of 14.08 Å (ref. 66). This subset of the data was subsequently aligned and classified in STOPGAP to generate a reference from the tomographic data with a Fourier shell correlation (FSC) value close to 1 at 40 Å template-matching resolution. Template matching with various GroEL$_{14}$ species was attempted, but never yielded an average of GroEL$_{14}$ with a resolution better than the template resolution. The data-derived references of all three different structures were used for an additional round of template matching on the complete dataset. Cross-correlation cut-off was chosen separately for every tomogram by visual inspection of the generated hits and comparison with the tomogram. To reduce the level of false-positive detection, a mask for the cytosol of the cell was first created using AMIRA (Thermo Fisher Scientific) and subsequently used to filter out hits outside of the cytosol. Putative particles were deliberately overpicked with low-resolution templates in the initial stage to avoid false-negative assignments.

This procedure yielded 176,408 initial subtomograms for the EL–ES$_1$ reference and 125,860 for the EL–ES$_2$ reference. These were then further aligned and classified separately in STOPGAP, each yielding classes containing both EL–ES$_1$ and EL–ES$_2$ particles. The combined number of particles contained in classes with emergent high-resolution features (Supplementary Fig. 1a) for the EL–ES$_1$ reference was 19,239, and 17,614 for the EL–ES$_2$ reference (Extended Data Fig. 1 and Supplementary Fig. 1b). Because both references pick up a subset of the other particles, the particles were then combined and duplicates removed. The resulting combined dataset was split by reference-free, three-dimensional classification in STOPGAP, resulting in a set of 17,598 EL–ES$_1$ and 11,213 EL–ES$_2$ complexes that were then independently refined.

This resulted in a resolution at the FSC cut-off of 0.143 following the application of symmetry at 11.6 Å for the EL–ES$_1$ complex (C7 symmetry) and 11.9 Å for the EL–ES$_2$ complex (D7 symmetry). Classification was performed using simulated annealing stochastic hill-climbing multireference alignment as previously described[67]. All classifications were done repeatedly with different, random initial starting sets of 250–500 subtomograms to generate the initial references. Only particles that ended up in the same class for all independent rounds of classifications were retained[67]. Further refinements with the established WARP, RELION, M pipeline were attempted but did not yield any further improvements. EL–ES$_1$ wide and narrow complexes were separated by classification with a focused, disk-shaped mask on the apical domains of the EL–ES$_1$ trans-ring. This resulted in 6,681 narrow complexes that were refined to a resolution of 13.5 Å, and 10,130 wide EL–ES$_1$ complexes refined to a resolution of 12.0 Å.

The EL$^+$ dataset was processed in the same way, but starting with the structures from the other datasets, low-pass filtered to 40 Å, as initial references for template matching. Template matching was then repeated once with structures generated by averaging a subset of particles from this dataset. To improve the resolution for model building, the dataset was exported to WARP[68] and angles and positions refined using RELION v.3.0.8 (ref. 69). This yielded a GroEL$_{14}$ structure at a global resolution of 13 Å. GroEL 14-mer particles were corefined for geometric distortions with ribosomes in M. The resulting GroEL 14-mer particles were exported for further alignment and classification in RELION. Classification was performed with a regularization parameter T of four and six classes for 25 iterations without angular search, resulting in a more homogeneous subset of 12,421 particles. These particles were again corefined in M for geometric distortions and per-particle defocus for contrast transfer function (CTF) estimation, resulting in a final structure with nominal resolution of 9.8 Å at 0.143 FSC cut-off.

Owing to their high molecular weight and density, ribosome template matching achieves a higher precision and recall. During initial rounds of classification in STOPGAP, because no false-positive particles were detected, all ribosomal hits from template matching were aligned first in STOPGAP at progressively lower binnings (bin4, bin2, bin1). The resulting particles were then exported to WARP using TOMOMAN. Subtomograms were reconstructed for RELION v.3.0.8 using WARP at a pixel size of 3.52 Å per pixel. An iterative approach with subtomogram alignment in RELION and tilt-series refinement in M[70] were performed until no further improvement in gold-standard FSC was obtained. This resulted in a final structure of the ribosome at a resolution of 8.6 Å for the combined 37 °C, HS and MetK datasets, and 6.3 Å for the EL$^+$ dataset, which was processed separately.

In vitro cryo-ET data for GroEL–GroES complexes were processed analogous to the in situ data, resulting in 39,518 initial hits for the EL–ES$_2$ template and 46,093 for the EL–ES$_1$ template, with both sets having a significant overlap. These were then further aligned and classified separately in STOPGAP, yielding 5,832 and 13,688 particles, respectively, following duplicate removal.

**Classification of SP occupancy of GroEL–GroES complexes in situ**
For the resolution of densities corresponding to substrate proteins in the GroEL–GroES chamber we first performed symmetry expansion around the C2 axis of the EL–ES$_2$ complexes and aligned the new set of GroEL–GroES chambers with the cis-ring of the EL–ES$_1$ complexes. The resulting subtomograms of the chambers were then denoised using TOPAZ's three-dimensional pretrained denoising function[71]. Because initial attempts to classify the interior of the chamber using STOPGAP multireference-based alignment showed only separation by missing wedge, the subtomograms were combined into 5,000 random bootstraps containing 250 random subtomograms each. These averages were then used to perform k-means clustering with two classes. Bootstraps from the resulting clusters were averaged and used as initial

start structures for multireference alignment in STOPGAP. For this, stochastic hill climbing was performed with a temperature factor of 10 for simulated annealing, followed by 40 iterations of multireference alignment with two classes and a mask around the interior of the chamber. This process was repeated five times. Only particles consistently assigned to the same classes were used for a final round of subtomogram averaging, resulting in one class showing weak diffuse density inside the chamber and a second showing strong density near the bottom. Attempts to further subdivide these two classes resulted only in separation based on missing wedge. Because it was not possible to resolve the C7 symmetry mismatch of the substrate and enclosing chamber, final averages were produced for all different biological conditions with C7 symmetry applied to increase the signal-to-noise ratio. The class showing a strong density near the bottom contained 12,255 subtomograms, the one showing only a weak diffuse density with 24,435 subtomograms for the combined 37 °C, HS and MetK datasets.

In vitro data were processed analogously. The resulting classes were then again split into EL–ES$_1$ and EL–ES$_2$ complexes corresponding to their substrate state and exported to WARP[68]. An additional round of alignments was performed in RELION for all different classes and complexes. A prior was set for all angles. Local search was performed with a sigma of 0.5 and search angle of 0.9°. The resulting particles were separately refined in M, correcting for geometrical distortions. Particles were again exported from M[70] and signal subtraction preformed in RELION of the trans-ring for EL–ES$_1$ and the opposing chambers for EL–ES$_2$. Based on their previous classification results in STOPGAP, the refined signal-subtracted, single-chamber complexes were combined in two groups resulting in 7,087 GroEL–GroES chambers containing an ordered SP and 14,371 that either contained a disordered SP or were empty. The resulting chambers were again locally refined in RELION using priors and a sigma on all angles, yielding a resolution of 9.4 Å for GroEL–GroES chambers containing ordered SP and 8.8 Å for the remaining chambers.

**Cryo-EM single-particle analysis of GroEL–GroES–MetK complexes**
For generation of substrate-bound GroEL–GroES complexes, 4 μM MetK was denatured in the presence of 1 μM GroEL (14-mer) in buffer A (20 mM MOPS-NaOH pH 7.4, 200 mM KCl, 10 mM MgCl$_2$, 5 mM DTT) containing 30 mM NaF and 5 mM BeSO$_4$ by first incubation of the mixture at 60 °C for 15 min and then cooling to 25 °C in a thermomixer (Eppendorf). The addition of 2 μM GroES (7-mer) and 1 mM ATP (pH 7.0) resulted in stable chaperonin complexes with encapsulated MetK[40]. Biochemical analysis of this preparation was performed by size exclusion chromatography on a Superdex 200 3.2/300 GL column. Fractions were analysed by SDS–PAGE electrophoresis (NuPAGE, Bis-Tris 4–12% gels), and MetK loading of GroEL–GroES complexes was estimated by mass spectrometry using intensity-based absolute quantification values[72]. For analysis by mass spectrometry, fractions F1 and F2 (Extended Data Fig. 5a) were analysed separately but intensities pooled for the determination of intensity-based absolute quantification ratios.

GroEL–GroES–MetK samples were concentrated tenfold by ultrafiltration using a 100 kDa Amicon centrifugal concentrator (Millipore) at room temperature. As a control, GroEL and GroES were treated identically in the absence of MetK. Before freezing, 1 μl of a n-octyl-β-D-glucopyranoside stock solution (87.5 mg ml$^{-1}$ in buffer A) was added per 50 μl of sample. For single-particle analysis and in vitro cryo-ET experiments, 4 μl of the sample was applied onto R 2/1 100 Holey carbon film Cu 200 mesh grids (Quantifoil) previously plasma cleaned for 30 s. This grid was blotted for 3.5 s at force 4 and plunge-frozen in a mixture of liquid ethane and propane cooled by liquid nitrogen using a Vitrobot Mark IV (Thermo Fisher Scientific) at 100% humidity and 4 °C.

Cryo-EM data for the EL–ES–MetK dataset were acquired using a FEI Titan Krios transmission electron microscope and SerialEM software[55].

Video frames were recorded at a nominal magnification of ×22,500 using a K3 direct electron detector (Gatan), with a total electron dose of around 55 electrons per Å$^2$ distributed over 30 frames at a calibrated physical pixel size of 1.09 Å. Micrographs were recorded within a defocus range of −0.5 to −3.0 μm.

On-the-fly image processing and CTF refinement of cryo-EM micrographs were carried out using the Focus software package[73]. Only micrographs that met the selection criteria (ice thickness under 1.05, drift 0.4 Å < $x$ < 70 Å, refined defocus 0.5 μm < $x$ < 5.5 μm, estimated CTF resolution under 6 Å) were retained. Micrograph frames were aligned using MotionCor2 (ref. 57), and the CTF for aligned frames was determined using GCTF[74].

The control dataset of GroEL–GroES complexes without MetK was acquired similarly but with a nominal magnification of ×29,000, resulting in a calibrated pixel size of 0.84 Å.

## Image processing, classification and refinement for single-particle analysis

From the resulting 8,945 micrographs of the GroEL–GroES–MetK dataset, 1,561,482 particles were picked using a trained crYOLO network[75] and extracted with RELION v.3.1.3 (ref. 69). An initial round of two-dimensional classification was performed and the remaining particles were passed into CryoSPARC[76] for further two-dimensional classification, ab initio model building, alignment and initial three-dimensional classification to separate EL–ES$_1$ from EL–ES$_2$ complexes. The remaining EL–ES$_2$ (659,866 particles) and EL–ES$_1$ (294,250 particles) complexes were then exported separately to RELION for additional alignment with imposed symmetry, CTF refinement and Bayesian polishing. For the EL–ES$_2$ complexes, symmetry expansion around the $C2$ axis was performed and the opposing half removed using RELION's signal subtraction.

The resulting asymmetric EL–ES$_1$ complexes were then classified further with CryoDRGN[77], resulting in a clean subset of 242,276 particles. The *trans*-rings of the EL–ES$_1$ complexes were classified in CryoSPARC using a focused mask on the apical domains of the *trans*-ring, resulting in 169,454 particles in the narrow conformation and 34,755 in the wide conformation. The resulting structures were refined in CryoSPARC under the application of $C7$ symmetry to a nominal resolution of 2.9 and 3.1 Å, respectively. For the analysis of the *cis*-chamber, all EL–ES$_1$ particles were pooled and the *trans*-ring was removed by signal subtraction in RELION.

The resulting GroES-bound, single-ring particles (1,562,002 particles) were then aligned to a common reference in RELION and exported to CryoSPARC for further alignment without imposed symmetry. The resulting mask and reference were reimported into RELION and used for an additional alignment step with the goal of aligning the asymmetric MetK substrate contained inside the chamber (Extended Data Fig. 5d). Subsequently a second round of signal subtraction was performed and the resulting particles, comprising only MetK density, were further subjected to three-dimensional classification without angular search in RELION. A subset of the resulting classes showed visible secondary structure elements in different orientations (Extended Data Fig. 5d). These classes were then combined and aligned into a single frame of reference in Matlab 2015b by manual rotation with the respective multiple of 360°/7 around the sevenfold symmetry axis. This was done by adding the corresponding increment to particle rotation angles in the particle table (.star file).

These folded MetK (fMetK) particles were then further locally aligned in CryoSPARC. An additional round of three-dimensional classification was performed followed by a final round of local alignment (322,800 particles), resulting in density for MetK at a resolution of 3.7 Å.

For the study of MetK contacts with the inner wall of GroEL–GroES chamber, we reverted the signal subtraction in RELION to generate single-ring GroEL–GroES–MetK particles for both the folded MetK and mixed population of chambers either containing disordered MetK or empty; both were refined and aligned in CryoSPARC. The subset containing a mixed population was additionally classified in CryoDRGN between the final alignment steps, resulting in a global resolution of 3.04 Å for the GroEL–GroES–MetK complex containing folded MetK and of 2.94 Å for the complex containing a mixed population of disordered MetK or empty chambers.

GroEL–GroES complexes without MetK were processed analogously but without Bayesian polishing and CTF refinement in RELION. Signal subtraction was performed in CryoSPARC; using 293,974 particles, this resulted in a map with a global resolution of 2.5 Å following the application of $C7$ symmetry.

Densities were visualized and rendered using ChimeraX[78,79].

## Model building and refinement

Model building was initiated by rigid-body fitting the GroEL subdomains, GroES and MetK from the crystal structures PDB 1SX3 (ref. 80), 5OPW[12] and 7LOO[42], respectively, into cryo-EM density, followed by manual editing using Coot[81]. The models were subsequently refined in real space with Phenix[82]. For the refinement of models against low-resolution data from STA, automatically generated restraints from reference structures such as PDB 8P4M (this study) were used. Residues with disordered sidechains were truncated at C-beta.

## Reporting summary

Further information on research design is available in the Nature Portfolio Reporting Summary linked to this article.

## Data availability

The mass spectrometry data have been deposited to the ProteomeXchange Consortium[83] via the PRIDE partner repository with the dataset identifier PXD042587. Model coordinates and electron density maps have been deposited to the wwPDB database under PDB/EMDB accession code nos. 8P4M/EMD-17418 (empty GroEL–GroES chamber), 8P4N/EMD-17420 (GroEL–GroES chamber with no or disordered MetK), 8P4O/EMD-17421/EMD-17422 (GroEL–GroES chamber with ordered MetK), 8QXS/EMD-18735 (EL–ES$_1$–MetK wide), 8QXT/EMD-18736 (EL–ES$_1$–MetK narrow), 8P4R/EMD-17426 (in situ EL–ES$_2$), 8QXU/EMD-18737 (in situ EL–ES$_1$ wide), 8QXV/EMD-18738 (in situ EL–ES$_1$ narrow) and 8P4P/EMD-17425 (in situ EL). Primary electron density maps have been deposited to the wwPDB database under EMDB accession code nos. EMD-17423 (in vitro GroEL–GroES chamber with no or disordered MetK), EMD-17424 (in vitro GroEL–GroES chamber with ordered MetK), EMD-17534 (empty EL–ES$_2$), EMD-17535 (empty EL–ES$_1$), EMD-17559 (GroEL–GroES chamber with no or disordered substrate), EMD-17560 (GroEL–GroES chamber with encapsulated, ordered substrate), EMD-17561 (70S ribosomes in 37 °C, HS and MetK *E. coli* cells), EMD-17562 (70S ribosomes in EL$^+$ *E. coli* cells), EMD-17563 (EL–ES$_1$ with encapsulated ordered MetK), EMD-17564 (EL–ES$_1$ with no or encapsulated disordered MetK), EMD-17565 (EL–ES$_2$ with two chambers with no or disordered MetK), EMD-17566 (EL–ES$_2$ with ordered MetK in one chamber and no or disordered MetK substrate in the other) and EMD-17567/EMD-17568/EMD-17569/EMD-17570/EMD-17571/EMD-17572/EMD-17573 (conformers 1–7 of EL–ES$_2$ with two encapsulated, ordered MetK). Because of their large file size, original cryo-ET imaging data are available from the corresponding author on request. Source data are provided with this paper.

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

**Acknowledgements** We thank S. Gärtner, R. Lange and N. Wischnewski for expert technical assistance. We thank L. Zhang and C. Sitron for help in developing the immunoprecipitation protocol and improving the text, respectively. This study used the infrastructure of the Department of Cell and Virus Structure at the MPI of Biochemistry. We thank J. Plitzko for valuable technical advice. Funding was provided by the German Research Foundation (Deutsche Forschungsgemeinschaft) under Germany's Excellence Strategy (EXC 2067/1-390729940) and SFB 1035, as well as the European Research Council (ERC Advanced Grant no. 101052783-INSITUFOLD) and the Ministry of Science and Culture of the State of Lower Saxony (74ZN1949).

**Author contributions** J.W. performed cryo-ET and cryo-EM single-particle analyses with help from W.W., F.B. and S.B. A.I.C. performed biochemical experiments and MS analysis together with J.W. and R.K. A.B. built the structural models and helped with data interpretation. F.U.H., W.B. and R.F.-B. designed the project and wrote the manuscript together with the other coauthors.

**Funding** Open access funding provided by Max Planck Society.

**Additional information**
**Correspondence and requests for materials** should be addressed to Wolfgang Baumeister, Ruben Fernandez-Busnadiego or F. Ulrich Hartl.

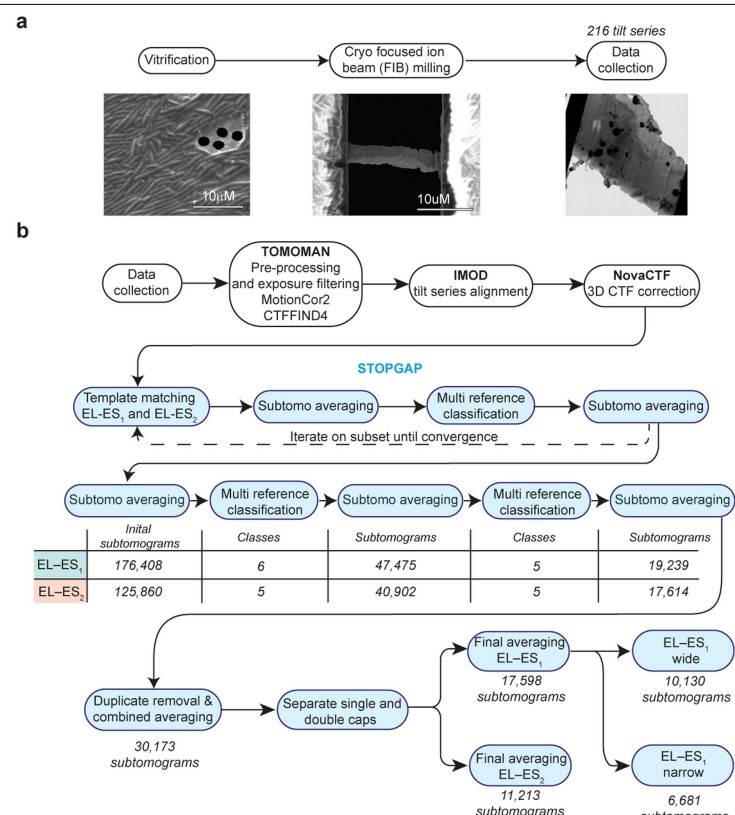

**Extended Data Fig. 1 | Cryo-ET and subtomogram averaging. (a)** Sample preparation for cryo-ET. *E. coli* cells were vitrified, thinned by cryo focussed ion beam (FIB) milling and tomograms aquired in a cryo-transmission electron microscope (TEM). Representative scanning electron micrograph of a sample before and after FIB milling is shown along with an overview of a lamella from a cryo-TEM (a total of 166 tomograms were acquired for 37 °C, HS and MetK combined). **(b)** Processing flowchart used for EL–ES$_1$ and EL–ES$_2$ subtomogram averaging in situ. The color of the box indicates whether the respective step was performed in STOPGAP (blue) or with the indicated program (white). See Methods for details.

**a**

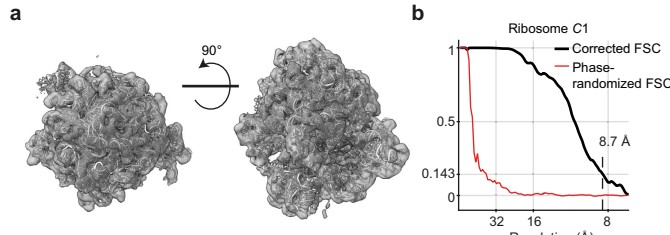

90°

**b**

Ribosome *C*1

— Corrected FSC
— Phase-randomized FSC

0.5

0.143

8.7 Å

32    16    8

Resolution (Å)

**c**

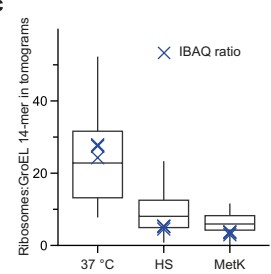

✕ IBAQ ratio

Ribosomes:GroEL 14-mer in tomograms

40

37 °C    HS    MetK

**Extended Data Fig. 2 | In situ structural analysis of 70 S ribosomes.**
(**a-b**) Subtomogram averaging of ribosomes. Ribosomes from three datasets
(37 °C, HS, MetK) were averaged and refined to a global resolution of 8.7 Å.
The resulting subtomogram structure with the superposed molecular model
(PDB code 4V4A[84]) in ribbon representation (a) and the corresponding FSC
curve (b) are shown. (**c**) Analysis of ribosome to GroEL 14-mer ratio in
tomograms (box plots; 37 °C n = 48, HS n = 58, MetK n = 60 tomograms) and
by MS using intensity-based absolute quantification (iBAQ) (blue crosses; n = 3
independent experiments). Box plots show median (center line), interquartile
range (IQR) (box edges) and 1.5 × IQR (whiskers). The MS measurements fall
mainly within the range of the first to third quartile of the tomography data,
indicating that most EL complexes were identified in situ.

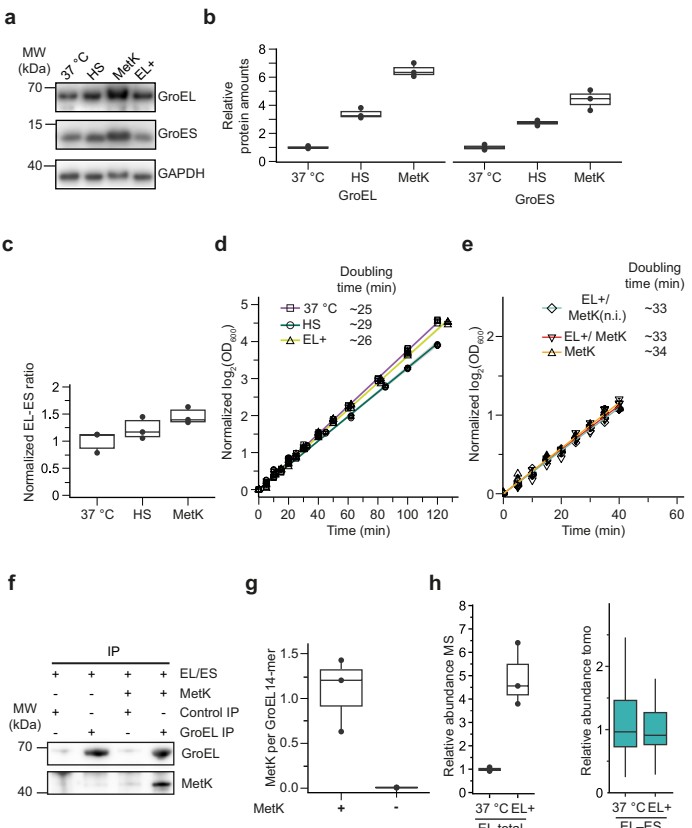

**Extended Data Fig. 3 | Biochemical analysis of GroEL/GroES levels and MetK binding.** (**a**) Representative immunoblot of GroEL and GroES for the different growth conditions analyzed (37 °C, HS, MetK and EL+). Glyceraldehyde-3-phosphate dehydrogenase (GAPDH) was used as loading control. (**b**) Quantification of GroEL and GroES levels by label-free mass spectrometry of cell lysates. The total amount of GroEL was quantified by label-free mass spectrometry using iBAQ. iBAQ values of GroEL and GroES of cells grown at 37 °C cells were set to 1 and used for normalization (n = 3 independent experiments). The horizontal line in the boxplots indicates the median value; boxes indicate upper and lower quartile and whisker caps the largest or smallest value within 1.5 times the interquartile range above the 75th percentile or below the 25th percentile, respectively. (**c**) Ratio of GroEL 14-mer and GroES 7-mer, based on the iBAQ values from (b). The GroEL:GroES ratio in wild-type cells at 37 °C was set to 1 and used for normalization. The differences between the groups were not statistically significant when compared with a 1-way ANOVA test. (**d**) Growth of *E. coli* BL21(DE3) at 37 °C, upon exposure to HS at 46 °C or upon ~4,5-fold overexpression of GroEL (EL+) at 37 °C. Data points are averages ± SD (n = 3, independent repeats). Growth curves were standardized to start at a $\log_2(OD_{600})$ value of 0. (**e**) Growth of transformed *E. coli* BL21(DE3) at

37 °C, upon sequential overexpression of GroEL/GroES and MetK (MetK cells) or upon ~4,5-fold overexpression of GroEL with subsequent overexpression of MetK (EL+/MetK) at 37 °C (see Methods). For comparison, the growth of the latter strain without induction (n.i.) of MetK (EL+/MetK(n.i.)) is shown. Data points are averages ± SD (n = 3, independent repeats). Growth curves were standardized to start at a $\log_2(OD_{600})$ value of 0. (**f**) Quantification of MetK bound to GroEL complexes in 37 °C and MetK overexpressing cells. Apyrase treatment was performed upon cell lysis to stop GroEL cycling. GroEL was immunoprecipitated (IP), followed by immunoblotting with antibodies against GroEL and MetK. Anti-lactalbumin antibodies were used as non-specific control. (**g**) Quantification of MetK:GroEL stoichiometry by MS in GroEL IPs from (d). The fraction of MetK per GroEL 14-mer was calculated based on iBAQ values (n = 3 independent experiments). Box plots show median (center line), interquartile range (IQR) (box edges) and 1.5 × IQR (whiskers). (**h**) Cellular abundance of GroEL in 37 °C and EL+ cells. The data was normalized to a median of 1 for 37 °C (n = 3 independent experiments). Boxplots are defined as in (g). (**i**) Cellular abundance of EL–ES$_1$ in 37 °C and EL+ cells. The abundance of EL–ES$_1$ relative to ribosomes in tomograms was calculated as a proxy for its cytosolic concentration and normalized to a median of 1 for 37 °C.

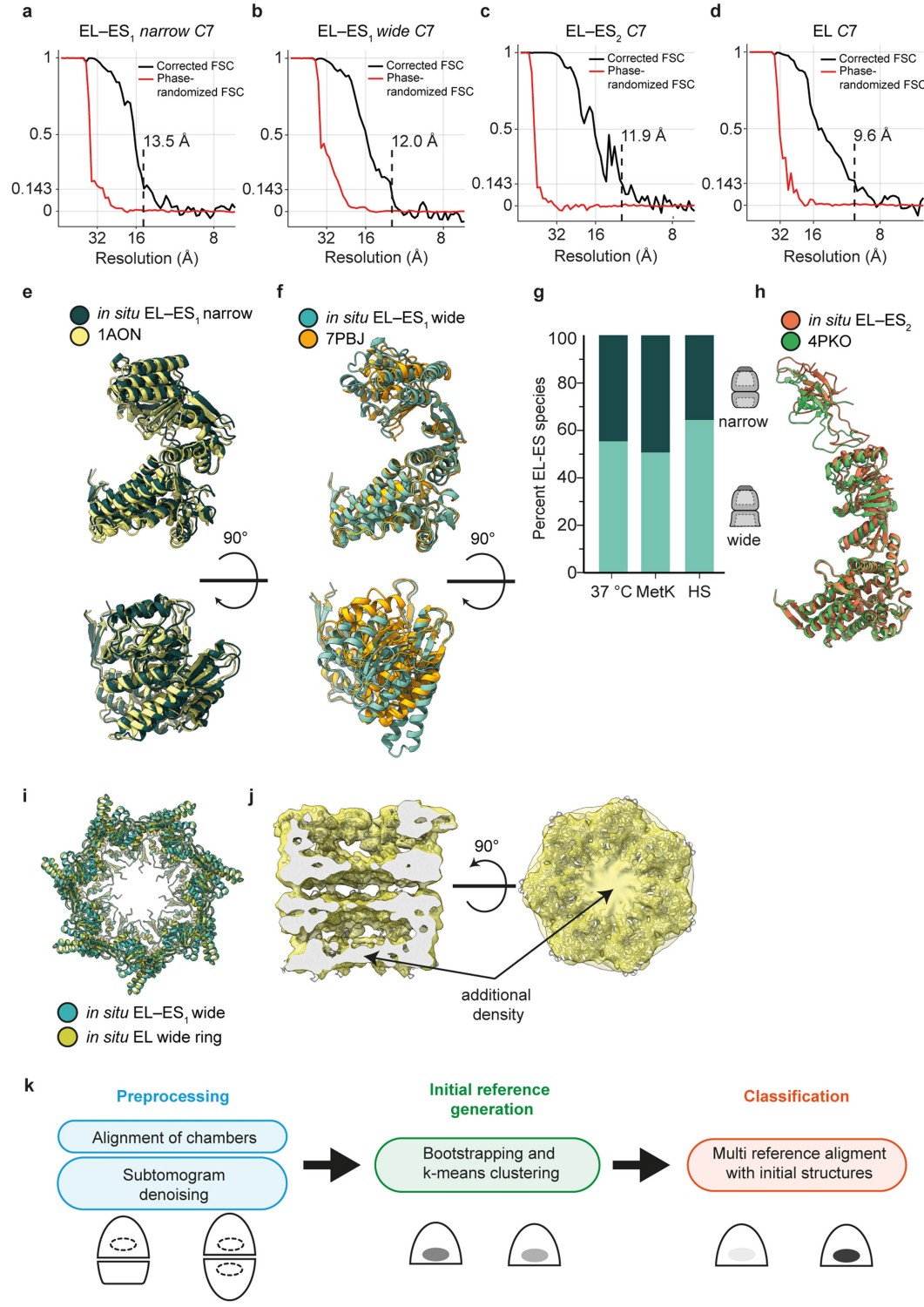

**a** EL–ES₁ *narrow C7*
**b** EL–ES₁ *wide C7*
**c** EL–ES₂ *C7*
**d** EL *C7*

— Corrected FSC
— Phase-randomized FSC

13.5 Å   12.0 Å   11.9 Å   9.6 Å

Resolution (Å)

**e** *in situ* EL–ES₁ narrow / 1AON
**f** *in situ* EL–ES₁ wide / 7PBJ
**g** Percent EL–ES species   37 °C MetK HS   narrow / wide
**h** *in situ* EL–ES₂ / 4PKO

90°

**i** *in situ* EL–ES₁ wide / *in situ* EL wide ring

**j** additional density   90°

**k**
**Preprocessing**
Alignment of chambers
Subtomogram denoising

**Initial reference generation**
Bootstrapping and k-means clustering

**Classification**
Multi reference alignment with initial structures

**Extended Data Fig. 4** | See next page for caption.

**Extended Data Fig. 4 | In situ structural analysis of GroEL complexes.**
(**a**-**d**) FSC curves for EL–ES$_1$ narrow (a), EL–ES$_1$ wide (b) and EL–ES$_2$ complexes (c), as well as EL (d) from Fig. 2a–d. The resolution at the 0.143 FSC cut-off is indicated. Note that free GroEL (EL) was only observed upon GroEL overexpression (EL+) and thus this structure was obtained from a separate data set. (**e**) Comparison of GroEL subunits in the *trans*-ring of the in situ structure of EL–ES$_1$ narrow (dark blue) and the crystal structure of GroEL·ADP$_7$–GroES$_7$ (yellow) (PDB 1AON[31]). Two orthogonal views of the superimposed models are shown. The models are depicted in ribbon representation. (**f**) Comparison of GroEL subunits in the *trans*-ring of the in situ structure of EL–ES$_1$ wide (light blue) and the cryoEM structure of EL–ES$_1$ in complex with 14 ADP molecules (orange) (PDB 7PBJ[32]), using the same representation as in (e). (**g**) Distribution of narrow and wide EL–ES$_1$ complexes at 37 °C, upon overexpression of GroEL, GroES and MetK at 37 °C (MetK cells) or upon exposure to HS at 46 °C (37 °C, n = 48; MetK, n = 60; HS, n = 58 tomograms). (**h**) Comparison of GroEL–GroES units of the in situ structure of EL–ES$_2$ (orange) and the crystal structure of EL–ES$_2$ in complex with 14 ADP·BeFx ligands (teal) (PDB 4PKO[33]). The models are depicted in ribbon representation. (**i**) Overlay of the rings in the wide conformation in the in situ structures of EL–ES$_1$ (teal) and the EL complex (yellow). The models are depicted in ribbon representation. (**j**) Cross section through the EL complex density. Additional density not accounted for by the molecular model is present in the more narrow ring at the SP binding sites, as shown in side and top view. There is no additional density at the given contour level in the opposing ring in the wide conformation. (**k**) Processing workflow of tomograms for analysis of encapsulated SP. To discern the SP states in GroEL–GroES chambers of EL–ES$_1$ and EL–ES$_2$ complexes, isolated chambers were aligned. After denoising the resulting subtomograms, initial structures for subsequent 3D classification were produced by bootstrapping and k means clustering. The resulting averages were used as starting structures for 3D classification (see Methods).

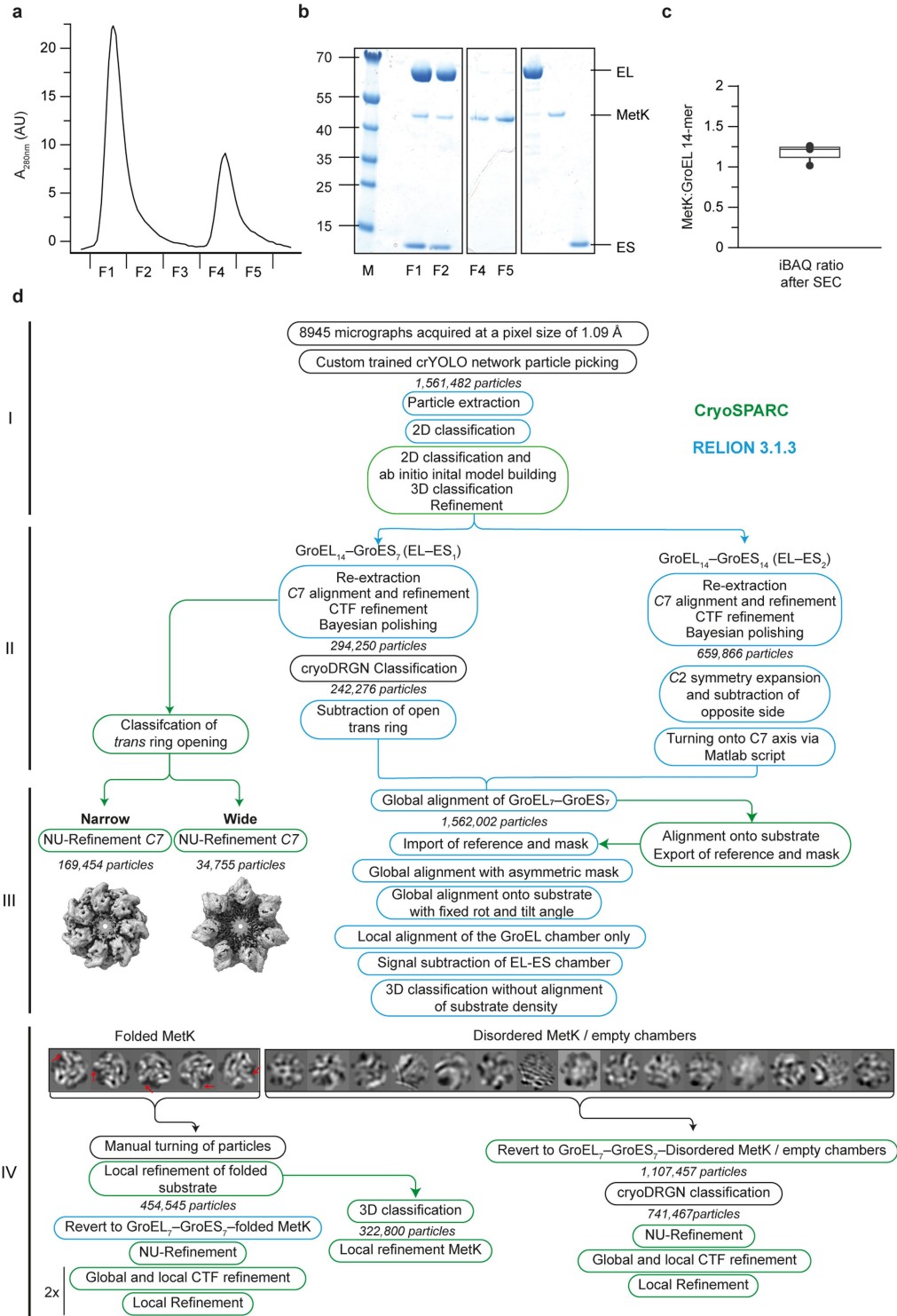

**Extended Data Fig. 5** | See next page for caption.

**Extended Data Fig. 5 | Preparation and cryo-EM analysis of the GroEL–GroES–MetK complex.** (**a-b**) Size exclusion chromatography of the stable GroEL–GroES–MetK complex prepared in the presence of ATP·BeF$_x$ (see Material and Methods). A representative chromatogram is shown in (a). Eluate fractions F1–F5 were analyzed by SDS-PAGE and Coomassie staining (b). Purified GroEL, MetK and GroES were analyzed for comparison. Fractions F1 and F2 contain the GroEL:ES-encapsulated MetK. (**c**) The complex was analyzed by MS and the ratio of MetK to the GroEL 14-mer calculated based on iBAQ values (n = 2 independent samples, each 3 technical repeats). Box plots show median (center line), interquartile range (IQR) (box edges) and 1.5 × IQR (whiskers). (**d**) Data processing workflow for single particle analysis of GroEL–GroES–MetK complexes. A flow diagram is shown. The color of the box borders indicate that the respective step was performed in CryoSPARC (green), RELION 3.1.3 (blue) or with the indicated program (black). After data collection, particle picking and initial 2D classification (I), EL–ES$_1$ and EL–ES$_2$ complexes were processed separately (II). GroEL–GroES chambers were extracted and combined for further processing. Subsequently, the GroEL–GroES density was subtracted and the chamber interiors separated by 3D classification without alignment (III). The picture row shows central slices of the resulting 3D class averages, which were used to separate folded MetK from disordered MetK or empty chambers (IV). Red arrows mark the MetK densities differing by 2π/7 rotation in the GroEL–GroES chambers. The other 3D class averages had no visible secondary structure elements. The GroEL–GroES–MetK chambers containing folded MetK were aligned and refined to a resolution of 3.0 Å. Local refinement of MetK after signal subtraction resulted in a 3.7 Å resolution map. The final resolution for GroEL–GroES chambers with disordered MetK or empty chambers was 2.9 Å. See Methods for details.

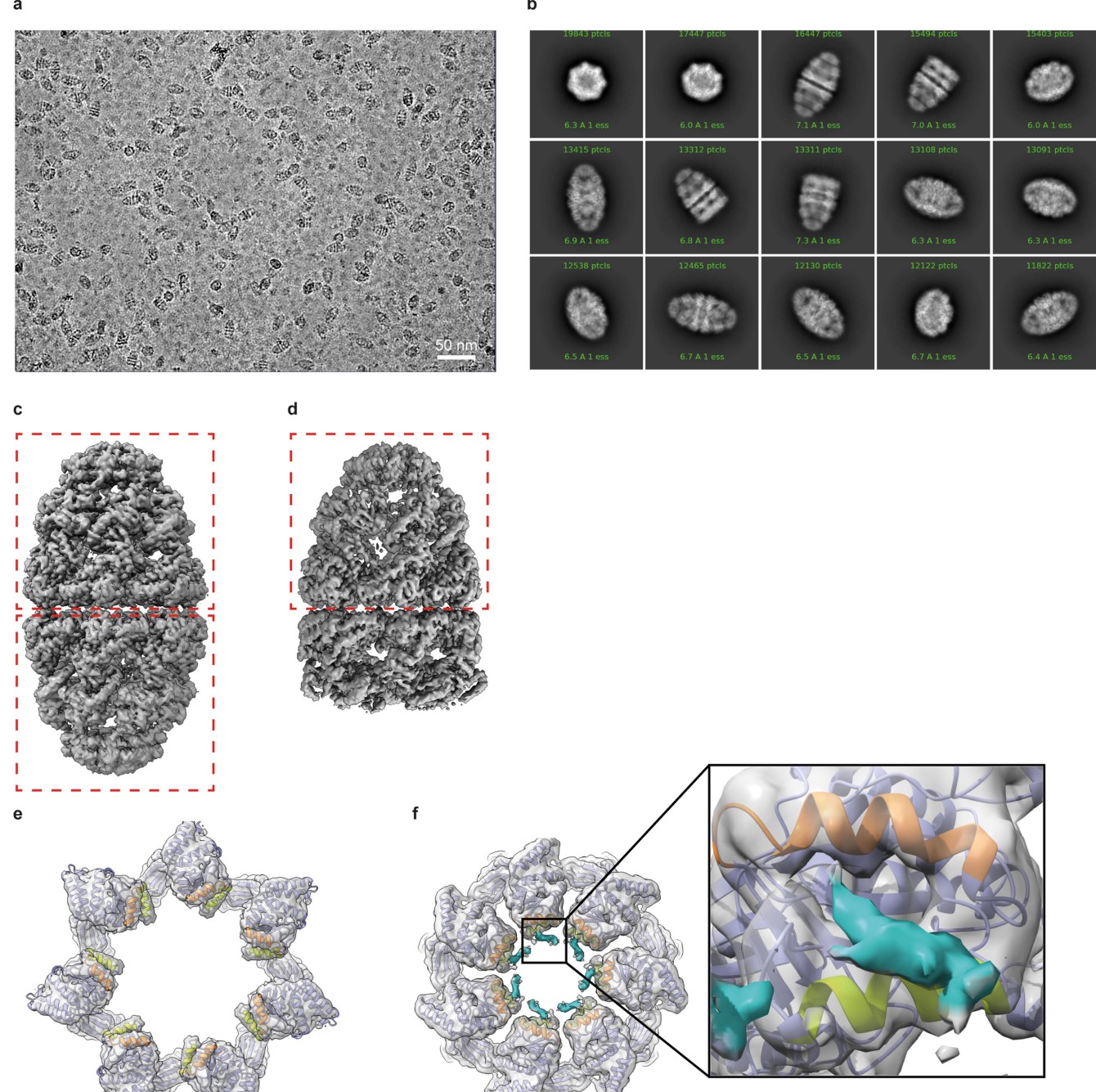

**Extended Data Fig. 6 | Single particle analysis of GroEL–GroES–MetK complexes.** (**a**) A representative micrograph of the GroEL–GroES–MetK sample at a magnification of 22,500-fold. (8,945 micrographs were used after on-the-fly preselection) (**b**) Corresponding 2D classes of particles selected for further refinement. (**c**, **d**) Surface representation of densities for MetK-containing EL–ES$_2$ (c) and EL–ES$_1$ (narrow conformation) complexes (d). The red boxes indicate the GroEL–GroES chambers that were processed further to solve the GroEL–GroES–MetK complex structure. (**e**, **f**) Bottom views of the densities and molecular models for the apical domains of the *trans*-ring in EL–ES$_1$ complexes with wide (e) and narrow (f) conformation. Surface views of the density are shown. The models are depicted in ribbon representation, with the substrate binding helices αH and αI highlighted in orange and yellow, respectively. Additional density in the narrow *trans*-ring not accounted for by the model – presumably from the substrate MetK – is high-lighted in teal (f). The insert shows one apical domain of the narrow *trans*-ring in detail.

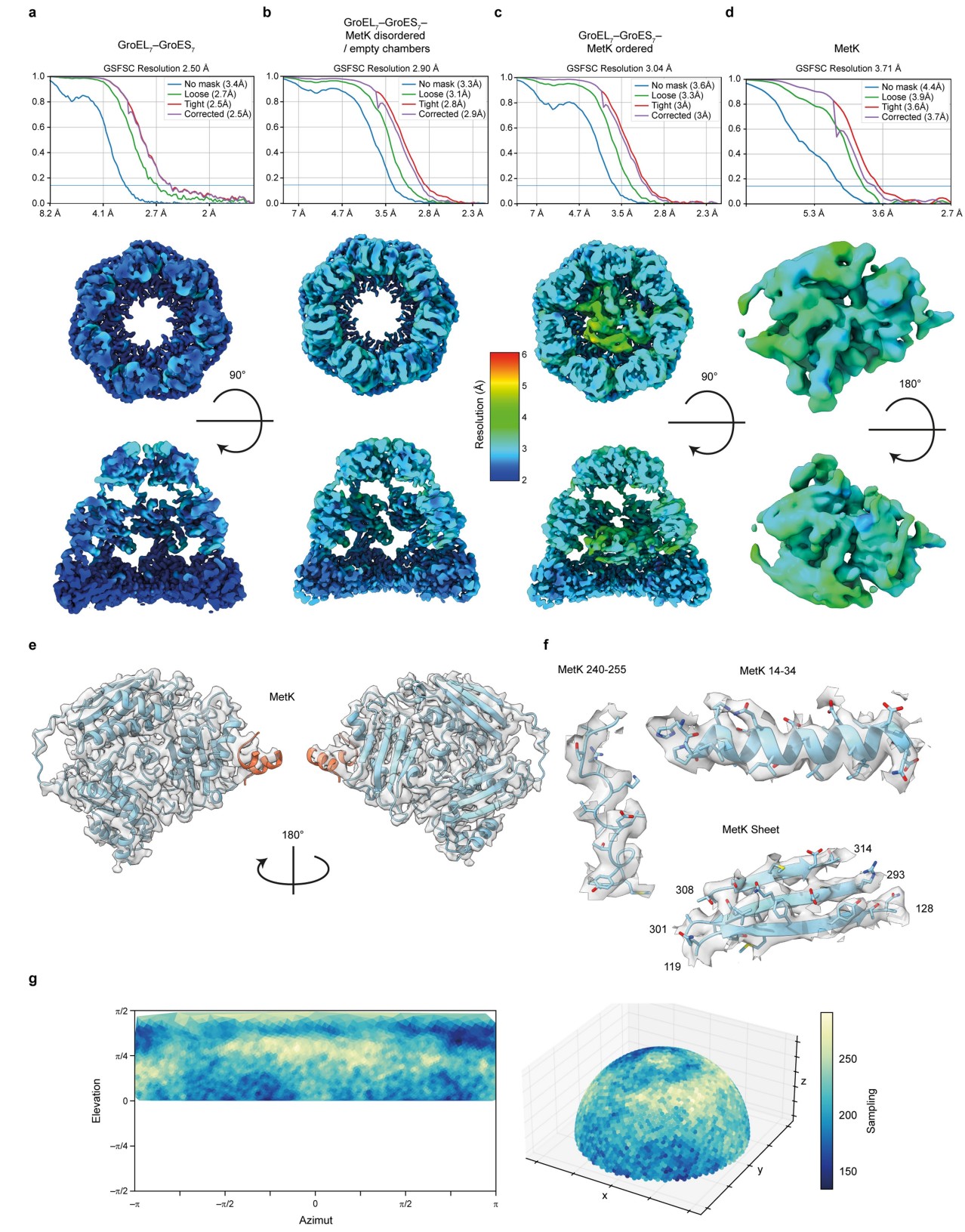

**Extended Data Fig. 7 | Resolution and density fit analysis of GroEL–GroES chambers and MetK.** (**a**-**d**) FSC curves (top) and local resolution maps (bottom) of the empty GroEL–GroES chamber (a), the GroEL–GroES chamber with disordered MetK or without substrate (b), the GroEL–GroES chamber with ordered MetK (c) and isolated MetK (d), respectively. The rainbow color gradient indicates the local resolution scale. (**e**, **f**) Cryo-EM density of MetK with superposed molecular model in ribbon representation. Two views for the entire protein are shown (e). Exemplary portions of the structure are shown below with side chains in stick representation (f). The respective residue ranges are indicated. (**g**) Angular sampling of MetK. Planar and spherical representation of the Fourier sampling of MetK depicted in (d). The color gradient from dark blue to pale yellow corresponds to the number of images in each bin. The sampling compensation factor was calculated to be 0.987 with values over 0.81, generally indicating adequate sampling[85,86]. Image was generated using the CryoSPARC Orientation Diagnostics job[76].

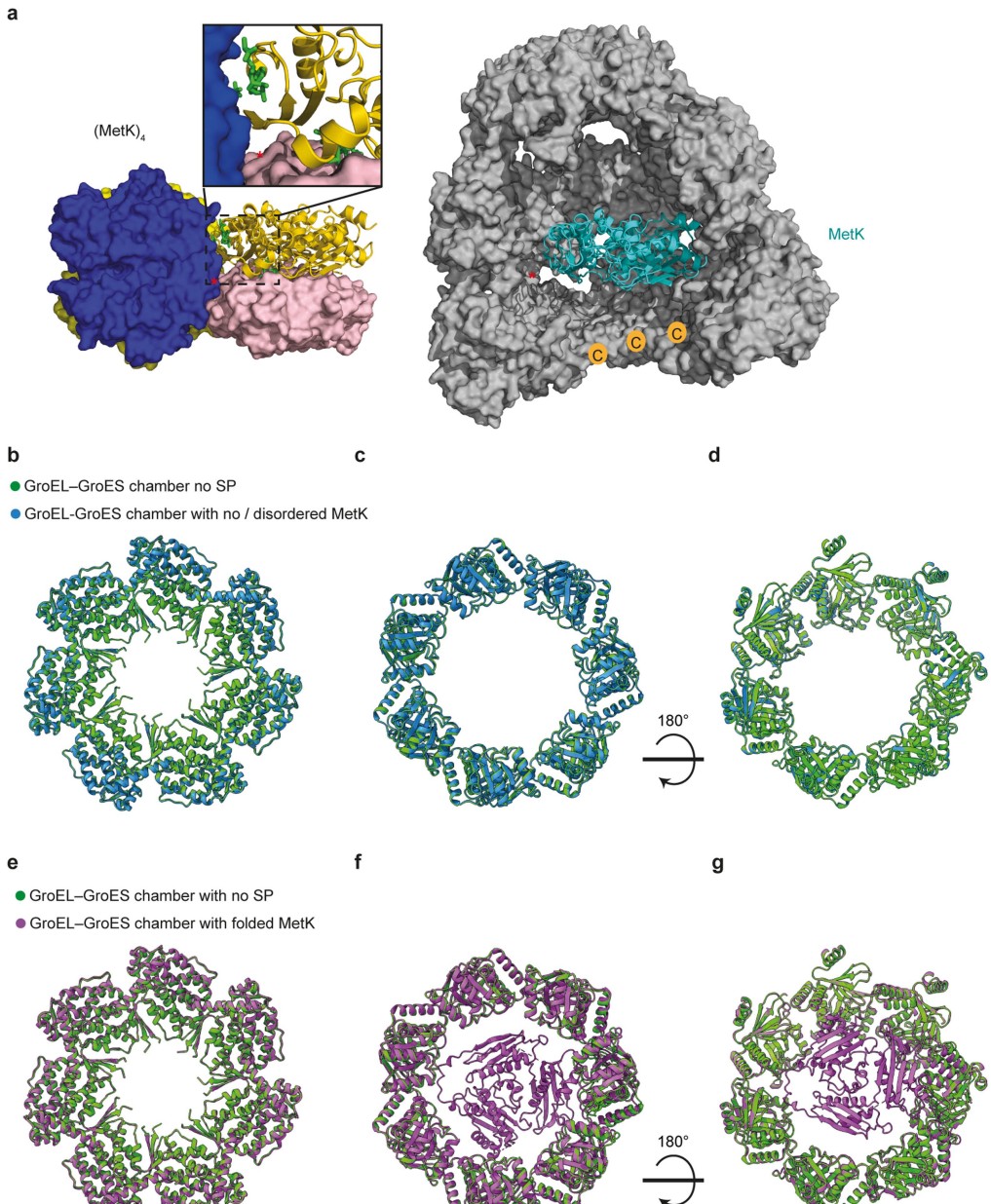

**Extended Data Fig. 8 | Comparison of isolated tetrameric MetK with GroEL–GroES-encapsulated MetK and the effect of MetK encapsulation on GroEL–GroES chambers.** (**a**) Overview of the crystal structure of the MetK tetramer (left; PDB 7LOO[42]). One subunit is shown in ribbon representation in gold, the other three as molecular surfaces in violet, blue and yellow, respectively. The insert highlights the location of the core loop, which is marked by a red asterisk. Bound ligands pyrophosphate and S-adenosylmethionine are shown as stick models in green. Cut-away view of GroEL–GroES encapsulated, folded MetK in the GroEL–GroES chamber in the same orientation (right). MetK is shown in ribbon representation (teal). The GroEL and GroES subunits are shown as molecular surfaces. The core loop of MetK is indicated by a red asterisk, and the last resolved residue Pro525 in the GroEL subunits is indicated with the letter C. The disordered C-terminal GGM repeats, GroEL residues 536–548, could easily reach the exposed MetK interface regions. (**b-d**) Overlay of the *C*7-symmetric model of the empty GroEL–GroES chamber (green) with the chamber containing either disordered MetK or no substrate (blue) at the level of the equatorial GroEL domains (b), the intermediate domains and the hinge regions between equatorial and intermediate domains (c and d). (**e-g**) Overlay of the *C*7-symmetric model of the empty GroEL–GroES chamber (green) with the chamber containing folded MetK (purple) at the level of the equatorial GroEL domains (e), the intermediate domains and the hinge regions between equatorial and intermediate domains (f and g).

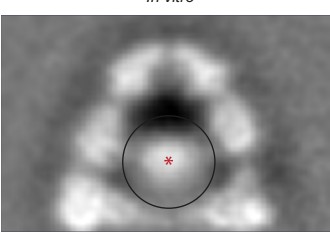 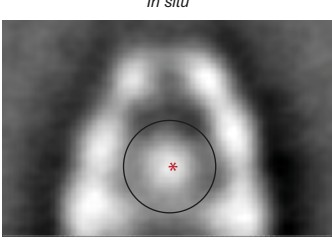

**Extended Data Fig. 9 | Analysis of the SP density in GroEL–GroES chamber subtomograms.** Central xy slices of the subtomogram averages of in vitro assembled GroEL–GroES chambers with ordered MetK (left) and class I in situ chambers (containing structured substrate; see Fig. 3b) from cells overexpressing GroEL/GroES and MetK (right), with gray values normalized to 2 standard deviations. The center of mass for the density within a spherical volume in the chamber indicated by a circle is depicted by a red asterisk. Within error, the centers of mass were identical (x, y, z in voxels: 65, 65, 51).

## Extended Data Table 1 | Cryo-ET and cryo-EM statistics and model validation

| | GroEL14-GroES14 (EMDB-17426, PDB 8P4R) | GroEL14-GroES7 wide (EMDB-18735, PDB 8QXU) | GroEL14-GroES7 narrow (EMDB-18736, PDB 8QXV) | GroEL14 (EMDB-17425, PDB 8P4P) | GroEL7-GroES7-MetK (EMDB-17421, PDB 8P4O) | GroEL7-GroES7-disordered MetK (EMDB-17420, PDB 8P4N) | GroEL7-GroES7 (EMDB-17418, PDB 8P4M) | GroEL14-GroES7-unresolved MetK wide (EMDB-18735, PDB 8QXS) | GroEL14-GroES7-unresolved MetK narrow (EMDB-18736, PDB 8QXT) |
|---|---|---|---|---|---|---|---|---|---|
| **Data collection and processing** | | | | | | | | | |
| Magnification | 42,000 | 42,000 | 42,000 | 42,000 | 22,500 | 22,500 | 29,000 | 29,000 | 29,000 |
| Voltage (kV) | 300 | 300 | 300 | 300 | 300 | 300 | 300 | 300 | 300 |
| Electron exposure ($e^-/Å^2$) | 120-160 | 120-160 | 120-160 | 120-160 | 55 | 55 | 55 | 55 | 55 |
| Defocus range (μm) | -3 – -5 | -3 – -5 | -3 – -5 | -2 – -4 | -0.5 – 3.0 | -0.5 – 3.0 | -0.5 – 3.0 | -0.5 – 3.0 | -0.5 – 3.0 |
| Pixel size (Å) | 3.52 | 3.52 | 3.52 | 3.02 | 1.09 | 1.09 | 0.84 | 0.84 | 0.84 |
| Symmetry imposed | *D7* | *C7* | *C7* | *C7* | *C1* | *C1* | *C7* | *C7* | *C7* |
| Initial particle images (no.) | 125,860 | 176,408 | 176,408 | 111,921 | 1,562,482 | 1,562,482 | 471,808 | 1,562,482 | 1,562,482 |
| Final particle images (no.) | 17,614 | 10,130 | 6,681 | 12,421 | 454,545 | 741,467 | 293,974 | 34,755 | 169,454 |
| Map resolution (Å) | 11.6 | 13.5 | 12 | 9.7 | 3.04 | 2.9 | 2.5 | 3.12 | 2.9 |
| FSC threshold | 0.143 | 0.143 | 0.143 | 0.143 | 0.143 | 0.143 | 0.143 | 0.143 | 0.143 |
| **Refinement** | | | | | | | | | |
| Initial models used (PDB code) | 8P4M | 1SX3, 8P4M | 1SX3, 8P4M | 1SX3, 4AB3 | 1SX3, 5OPX, 7LOO | 1SX3, 5OPX | 1SX3, 5OPX | 1SX3, 5OPX | 1SX3, 5OPX |
| Model resolution (Å) | 12 | 13.5 | 12 | 9.6 | 3.04 | 2.90 | 2.50 | 3.12 | 2.9 |
| FSC threshold | 0.143 | 0.143 | 0.143 | 0.143 | 0.143 | 0.143 | 0.143 | 0.143 | 0.143 |
| Model resolution range (Å) | 2.7 – 40 | 2.7 – 40 | 2.7 – 40 | 2.7 – 40 | 2.7 – 40 | 2.7 – 40 | 2.7 – 40 | 2.7 – 40 | 2.7 – 40 |
| Map sharpening B factor ($Å^2$) | -800 | -800 | -800 | -800 | -109 | -120 | -115 | -95 | -115 |
| **Model composition** | | | | | | | | | |
| Non-hydrogen atoms | 64,400 | 59,458 | 59,521 | 54,509 | 34,605 | 32,130 | 32,200 | 59,269 | 59,311 |
| Protein residues | 8,666 | 8,008 | 8,008 | 7,343 | 4,712 | 4,333 | 4,333 | 8,008 | 8,008 |
| Ligands | K: 14 MG: 14 ATP: 14 | K: 14 MG: 14 ATP: 7 ADP: 7 | K: 14 MG: 14 ATP: 7 ADP: 7 | K: 14 MG: 14 ATP: 7 ADP: 7 | K: 7 MG: 7 BEF: 7 ADP: 7 | K: 7 MG: 7 BEF: 7 ADP: 7 | K: 7 MG: 7 BEF: 7 ADP: 7 | K: 14 MG: 14 BEF: 7 ADP: 14 | K: 14 MG: 14 BEF: 7 ADP: 14 |
| **B factors ($Å^2$)** | | | | | | | | | |
| Protein | 992.61 | 977.57 | 960.27 | 544.95 | 63.01 | 29.51 | 53.57 | 91.46 | 56.14 |
| Ligand | 999.99 | 999.99 | 999.99 | 322.13 | 42.05 | 9.80 | 41.07 | 35.04 | 28.21 |
| Water | 972.36 | 979.14 | 894.04 | 299.45 | 30.98 | 9.72 | 29.92 | 35.62 | 24.47 |
| **R.m.s. deviations** | | | | | | | | | |
| Bond lengths (Å) | 0.002 | 0.004 | 0.004 | 0.004 | 0.002 | 0.004 | 0.004 | 0.002 | 0.005 |
| Bond angles (°) | 0.558 | 0.741 | 0.730 | 0.672 | 0.519 | 0.619 | 0.595 | 0.564 | 0.646 |
| **Validation** | | | | | | | | | |
| MolProbity score | 1.68 | 2.27 | 2.02 | 2.22 | 1.34 | 1.31 | 1.34 | 1.35 | 1.31 |
| Clashscore | 14.85 | 34.98 | 29.38 | 28.04 | 6.18 | 5.70 | 6.13 | 6.32 | 5.72 |
| Poor rotamers (%) | 0.00 | 0.00 | 0.00 | 0.00 | 0.06 | 0.06 | 0.81 | 0.00 | 0.00 |
| **Ramachandran plot** | | | | | | | | | |
| Favored (%) | 98.69 | 96.32 | 97.72 | 95.82 | 98.95 | 98.72 | 98.72 | 98.47 | 98.28 |
| Allowed (%) | 1.31 | 3.68 | 2.28 | 4.18 | 1.05 | 1.28 | 1.28 | 1.53 | 1.72 |
| Disallowed (%) | 0.00 | 0.00 | 0.00 | 0.00 | 0.00 | 0.00 | 0.00 | 0.00 | 0.00 |
| **Model-map CC values** | | | | | | | | | |
| $CC_{volume}$ | 0.84 | 0.79 | 0.71 | 0.72 | 0.77 | 0.80 | 0.84 | 0.81 | 0.79 |
| $CC_{mask}$ | 0.88 | 0.84 | 0.75 | 0.75 | 0.80 | 0.82 | 0.86 | 0.83 | 0.81 |
| $CC_{peaks}$ | 0.85 | 0.80 | 0.65 | 0.66 | 0.70 | 0.74 | 0.74 | 0.75 | 0.72 |

| | F. Ulrich Hartl |
| | Wolfgang Baumeister |

# Reporting Summary

## Statistics

For all statistical analyses, confirm that the following items are present in the figure legend, table legend, main text, or Methods section.

| n/a | Confirmed | |
|---|---|---|
| ☐ | ☒ | The exact sample size (*n*) for each experimental group/condition, given as a discrete number and unit of measurement |
| ☐ | ☒ | A statement on whether measurements were taken from distinct samples or whether the same sample was measured repeatedly |
| ☐ | ☒ | The statistical test(s) used AND whether they are one- or two-sided<br>*Only common tests should be described solely by name; describe more complex techniques in the Methods section.* |
| ☒ | ☐ | A description of all covariates tested |
| ☒ | ☐ | A description of any assumptions or corrections, such as tests of normality and adjustment for multiple comparisons |
| ☐ | ☒ | A full description of the statistical parameters including central tendency (e.g. means) or other basic estimates (e.g. regression coefficient) AND variation (e.g. standard deviation) or associated estimates of uncertainty (e.g. confidence intervals) |
| ☐ | ☒ | For null hypothesis testing, the test statistic (e.g. *F*, *t*, *r*) with confidence intervals, effect sizes, degrees of freedom and *P* value noted<br>*Give P values as exact values whenever suitable.* |
| ☒ | ☐ | For Bayesian analysis, information on the choice of priors and Markov chain Monte Carlo settings |
| ☒ | ☐ | For hierarchical and complex designs, identification of the appropriate level for tests and full reporting of outcomes |
| ☒ | ☐ | Estimates of effect sizes (e.g. Cohen's *d*, Pearson's *r*), indicating how they were calculated |

*Our web collection on statistics for biologists contains articles on many of the points above.*

## Software and code

Policy information about availability of computer code

| Data collection | Tomograms of 37 °C, MetK, HS E. coli cells were recorded on FEI Titan Krios transmission electron microscopes with SerialEM 3.9.056 software. Tomograms of EL+ E. coli cells were collected on a FEI Titan Krios transmission electron microscope with TEM Tomography 5 software (Thermo Fisher Scientific). CryoEM data for single particle averaging were recorded on a FEI Titan Krios transmission electron microscope with SerialEM software. On-the-fly image processing and contrast transfer function (CTF) refinement of the cryo-EM micrographs were carried out using the Focus 1.1 software package.<br>Mass spectrometry data were aquired on a Q-Exactive HF mass spectrometer (Thermo) with Xcalibur 4.0 and Q-Exactive HF-Orbitrap MS 2.7 software. |
|---|---|
| Data analysis | Cryo Electron Microscopy data was analysed using crYOLO 1.5.3, RELION 3.1.3 and CryoSPARC 2.0-3.3.2 as well as CryoDRGN 0.3.1-1.0.x. For display and initial analysis ChimeraX 1.4 and Chimera 1.13.1 was used. Model building was performed using Coot 0.9.4.1. Real-space refinement of molecular models was performed with Phenix 1.19.2.<br>Electron tomography data was analyzed with a combination of the following software packages: TOMOMAN 0.6, CTFfind 4.1.14, Stopgap 0.7.1, NovaCTF, MotionCor2 1.4.0, IMOD 4.11.1, Amira 2021.2, TomoSegMemTV 1.0, Relion 3.0.8, Warp 1.0.9 and M 1.0.9 Visualisation was performed with either IMOD 4.11.1 or ChimeraX 1.4.<br>Mass spectrometry data were analysed with MaxQuant 2.2.0.0.<br>Statistical analysis was performed with the R Studio 4.4.1 software package. |

For manuscripts utilizing custom algorithms or software that are central to the research but not yet described in published literature, software must be made available to editors and reviewers. We strongly encourage code deposition in a community repository (e.g. GitHub). See the Nature Portfolio guidelines for submitting code & software for further information.

## Data

The mass spectrometry data have been deposited to the ProteomeXchange Consortium via the PRIDE partner repository with the dataset identifier PXD042587 (https://www.ebi.ac.uk/pride/archive/projects/PXD042587). Model coordinates and electron density maps have been deposited to the wwPDB database under PDB/EMDB accession codes 8P4M/EMD-17418 (empty GroEL:ES chamber), 8P4N/EMD-17420 (GroEL:ES chamber with no or disordered MetK), 8P4O/EMD-17421/EMD-17422 (GroEL:ES chamber with ordered MetK), 8QXS/EMD-18735 (EL:ES1:MetK wide), 8QXT/EMD-18736 (EL:ES1:MetK narrow), 8P4R/EMD-17426 (in situ EL:ES2), 8QXU/EMD-18737 (in situ EL:ES1 wide), 8QXV/EMD-18738 (in situ EL:ES1 narrow) and 8P4P/EMD-17425 (in situ EL), respectively. Primary electron density maps have been deposited to the wwPDB database under EMDB accession codes EMD-17423 (in vitro GroEL:ES chamber with no or disordered MetK), EMD-17424 (in vitro GroEL:ES chamber with ordered MetK), EMD-17534 (empty EL:ES2), EMD-17535 (empty EL:ES1), EMD-17559 (GroEL:ES chamber with no or disordered substrate), EMD-17560 (GroEL:ES chamber with encapsulated, ordered substrate), EMD-17561 (70S ribosomes in 37 °C, HS and MetK E. coli cells), EMD-17562 (70S ribosomes in EL+ E. coli cells), EMD-17563 (EL:ES1 with encapsulated ordered MetK), EMD-17564 (EL:ES1 with no or encapsulated disordered MetK), EMD-17565 (EL:ES2 with two chambers with no or disordered MetK), EMD-17566 (EL:ES2 with ordered MetK in one chamber and no or disordered MetK substrate in the other chamber) and EMD-17567/EMD-17568/EMD-17569/EMD-17570/EMD-17571/EMD-17572/EMD-17573 (Conformers 1-7 of EL-ES2 with two encapsulated, ordered MetK). Because of their large file sizes, original cryo-ET imaging data are available from the corresponding author upon request. Source data to Fig. 1c and 3c and to Extended Data Fig. 2c, 3, 4g and 5 are provided with this paper.

## Research involving human participants, their data, or biological material

| | |
|---|---|
| Reporting on sex and gender | Not applicable. |
| Reporting on race, ethnicity, or other socially relevant groupings | Not applicable. |
| Population characteristics | Not applicable. |
| Recruitment | Not applicable. |
| Ethics oversight | Not applicable. |

Note that full information on the approval of the study protocol must also be provided in the manuscript.

# Field-specific reporting

Please select the one below that is the best fit for your research. If you are not sure, read the appropriate sections before making your selection.

☒ Life sciences  ☐ Behavioural & social sciences  ☐ Ecological, evolutionary & environmental sciences

For a reference copy of the document with all sections, see nature.com/documents/nr-reporting-summary-flat.pdf

# Life sciences study design

All studies must disclose on these points even when the disclosure is negative.

| | |
|---|---|
| Sample size | The amount of tomographic data acquired and used in this study was limited by the availability of microscopy time. For 37°C, HS and MetK at least 3 different biological replicates with several tomograms from different cells were acquired and analysed. For MetK only two independent biological replicates were used for data collection on dozens of different cells.<br>For cryoET a set of 48, 58, 60 and 64 curated tomograms for 37 °C, HS, MetK and EL+ were analyzed.<br>One in vitro GroEL:GroES:MetK samples was used for cryo-ET, resulting in 20 tomograms.<br>Three liquid culture samples, each of 37 °C, HS and EL+ E. coli cells were used for growth analysis.<br>Three liquid culture samples, each of MetK, EL+/MetK and EL+/MetK(n.i.) E. coli cells were used for growth analysis.<br>Three liquid culture samples, each of 37 °C, HS, MetK and EL+ E. coli cells were used for SDS-PAGE analysis of GroEL, GroES and GAPDH, for mass-spectroscopic analysis of ribosomal proteins, GroEL, GroES and MetK, and for immunoprecipitation and SDS-PAGE analysis of GroEL and MetK.<br>To determine via mass-spectroscopic analysis the stoichiometry of GroEL and MetK in GroEL:GroES:MetK complexes that were reconstituted in vitro, two independent samples were analyzed.<br>For cryoEM single-particle averaging of GroEL:GroES and GroEL:GroES:MetK complexes, one cryo-EM grid sample each was used for data collection and analysis. |

|  | No sample size calculation was performed. At least three independent experiments were performed in all cases as per commonly accepted standards of the field and to enable statistical analysis. |
|---|---|
| Data exclusions | For cryoEM, micrographs with an ice thickness score < 1.05, drift 0.4 Å < x < 70 Å, refined defocus 0.5 μm < x < 5.5 μm, estimated CTF resolution < 6 Å were selected on the fly, resulting in a dataset of 8,945 micrographs for the GroEL:ES:MetK sample. After 2D classification, only classes with visible secondary structure and a high estimated resolution were retained.<br>For cryoET, tilt series showing reflections from non-vitreous ice were discarded immediately after collection. Tilt series that had low aligment scores (residual error > 0.8) were discarded after reconstruction in IMOD. Particles that did not repeatedly end up in classes showing clear intermediate resolution features were discarded after 3D classification. This resulted in a selection of 19,239 subtomograms out of 176,408 starting subtomograms after template matching for EL:ES1 and 17,614 subtomograms out of 125,860 for EL:ES2, respectively. |
| Replication | 48, 58, 60 and 64 cryo-ET tomograms were used from 37 °C, MetK, HS and EL+ E. coli cells, respectively.<br>Biochemical experiments to establish growth rates, to quantify proteins by label-free mass spectrometry, to analyze protein expression by SDS-PAGE and Western blotting, and to test MetK association with GroEL by immune precipitation and label-free mass spectrometry were replicated 3 times. All attempts of replication were successful. |
| Randomization | Randomization did not apply to this study, because there was no assignment of data points to distinct group. The only relevant randomization is that of determining random half-sets of particles for resolution assessment in cryo-EM and cryo-ET reconstructions. Randomization in half sets for FSC determination was done internally in RELION, CryoSPARC and Stopgap 0.71, respectively. |
| Blinding | No blinding was applied since data collection and analysis were not strongly dependent on subjective interpretation of the data. The findings are supported by quantitative measurements and statistical analysis when relevant. |

# Reporting for specific materials, systems and methods

We require information from authors about some types of materials, experimental systems and methods used in many studies. Here, indicate whether each material, system or method listed is relevant to your study. If you are not sure if a list item applies to your research, read the appropriate section before selecting a response.

## Materials & experimental systems

| n/a | Involved in the study |
|---|---|
| ☐ | ☒ Antibodies |
| ☒ | ☐ Eukaryotic cell lines |
| ☒ | ☐ Palaeontology and archaeology |
| ☒ | ☐ Animals and other organisms |
| ☒ | ☐ Clinical data |
| ☒ | ☐ Dual use research of concern |
| ☒ | ☐ Plants |

## Methods

| n/a | Involved in the study |
|---|---|
| ☒ | ☐ ChIP-seq |
| ☒ | ☐ Flow cytometry |
| ☒ | ☐ MRI-based neuroimaging |

## Antibodies

| Antibodies used | The polyclonal antisera against GroEL (1:10,000), GroES (1:10,000), MetK (1:5000) and GAPDH (1:10,000) were produced in house. Rabbit antiserum against alpha-lactalbumin was a product of East Acres Biologicals, Southbridge, MA immunization service (1992). Anti-Rabbit IgG (whole molecule)–Peroxidase antibody produced in goat (Sigma-Aldrich A9169, 1:10,000). |
|---|---|
| Validation | Inhouse antibodies were validated by binding to purified protein using Western blot.<br>Western blot application of GroEL and MetK antisera was demonstrated in Kerner, Michael J., et al. "Proteome-wide analysis of chaperonin-dependent protein folding in Escherichia coli." Cell 122.2 (2005): 209-220. |

## Plants

| Seed stocks | *Report on the source of all seed stocks or other plant material used. If applicable, state the seed stock centre and catalogue number. If plant specimens were collected from the field, describe the collection location, date and sampling procedures.* |
|---|---|
| Novel plant genotypes | *Describe the methods by which all novel plant genotypes were produced. This includes those generated by transgenic approaches, gene editing, chemical/radiation-based mutagenesis and hybridization. For transgenic lines, describe the transformation method, the number of independent lines analyzed and the generation upon which experiments were performed. For gene-edited lines, describe the editor used, the endogenous sequence targeted for editing, the targeting guide RNA sequence (if applicable) and how the editor was applied.* |
| Authentication | *Describe any authentication procedures for each seed stock used or novel genotype generated. Describe any experiments used to assess the effect of a mutation and, where applicable, how potential secondary effects (e.g. second site T-DNA insertions, mosiacism, off-target gene editing) were examined.* |

