## [Peer Review File · Nature]

Manuscript Title: Visualizing chaperonin function in situ by cryo-electron tomography

Reviewer Comments & Author Rebuttals

Reviewer Reports on the Initial Version:

Referees' comments:

Referee #1 (Remarks to the Author):

In this pioneering work, Hartl and collaborators use cryo-ET to examine the folding cycle of the GroEL/S chaperonin in vivo. Both the type of questions asked and the interpretation of results benefit from Hartl's intimate knowledge of the in vitro GroEL folding literature, much of it contributed by his own laboratory. Additionally, the types of comparisons made and state of the art data analysis solidified significant conclusions--e.g. comparing groEL number to ribosome number, and undertaking specific in vitro experiments to allow extrapolation between data sets to establish which complex contain "folded substrate" in the GroEL complexes in vivo. I believe that this work is the leading edge of a new series of investigations that compare previously determined in vitro paradigms to actual in vivo function of cellular machines. This makes it very important to consider carefully how the work is presented. The comments below are suggestions when you revise the ms.

1. The basic framing of the manuscript is that there are many unsolved issues from the in vitro literature that this in vivo study will clarify. Are you sure that this is the best framing for your pioneering work, rather than documenting and rationalizing where the in vitro and in vivo results converge and differ. For example, although there have certainly been results in the literature supporting function either as a "bullet" or a "football", most of these investigations were many years ago, and at least my interpretation is that currently, most people believe that both types exist, and there is some evidence that the asymmetric football form is preferred. Note that if you agree with my impression of the current literature, I do not feel that it detracts at all from the significance of your results, but instead enhances them. For example, your finding of the distribution of complexes in exponentially growing cells is very interesting. It shows that essentially all GroEL's in vivo have the potential to be active, and that even though there is a 2:1 ratio of GroES to L, not all complexes are footballs, a fact explained later by the slow binding of the 2nd groEL. By focusing the ms. on differences instead of novelties, the authors may actually lose an opportunity to explain the significance of their data.

2. The author's route to identifying masses in GroEL as folded substrates is a strength of the ms (Fig. 3,4). This point could be further validated by comparing in vivo and in vitro cryoET of MetK for a viable folding defective GroEL mutant such as the GFP-enhanced variants described in Weissman's Cell 2002 111: 1027-39 paper to see if they also track in vivo and in vitro.

Minor points.

1. L 108-110. Incorrect as stated--the authors neglect to consider that there is extensive regulation of the amount of proteins in an operon to maintain preferred ratio of subunits (Cell. 2014 Apr 24;157(3):624-35. doi: 10.1016/j.cell.2014.02.033.). However, in this case, according to the data in that ms, GroES and GroEL are produced in a 1:1 ratio, but not because it is a consequence of operon organization--rather the cell has decided that this is optimal.

2. L117-118. EL:ES1 complexes predominate in ALL conditions tested.

3. Growth experiments.

(1) Were cells in all experiments harvested at a comparable OD? This is important because cell steady state constantly changes during growth in LB, and this could reflect EL/ES state. Could the change in EL/ES after heat shock actually sampling of culture closer to stationary phase than in the other growth conditions. This should be clarified in the methods

(2) It would be nice to have growth data for overexpression of GroEL/S + MetK to determine whether cells are in a significantly impaired state. Also note that ribosome number changes with growth rate.

4. The authors state that after GroEL overexpression, the ratio of GroEL/GroES1 to ribosomes is comparable to that in wt cells. Does that mean that there is a 40% reduction in the amount of total GroEL/S complexes? It would be interesting to examine the chaperonin properties (ie =Ext. fig 4g), and cellular response MetK overproduction--substrate folding properties of this strain (ie. Figs 3,4) to see whether any of the measurable parameters of the reaction are perturbed.

Referee #2 (Remarks to the Author):

Wagner et al. present a comprehensive study that reveals distinct conformations of the bacterial chaperonin GroEL and its cofactor GroES in their native cellular environment by combining cryo-focused ion beam (cryo-EIB) milling, cryo-electron tomography (cryo-ET), and advanced subtomogram analysis. The authors show evidence that the GroEL/ES complexes co-exist in asymmetrical (EL:ES1) and symmetrical (EL:ES2) forms in vivo under different growth conditions including heat stress and high level of substrate proteins. Furthermore, the authors determine in-situ structures of the GroEL/ES complexes together with folded substrate MetK. The in-situ structures are validated by high-resolution cryo-EM structures in vitro. Overall, the manuscript not only provides the most detailed view of the GroEL/ES assisted protein folding cycle in vivo, but also argues somewhat convincingly that both asymmetrical and symmetrical forms of the GroEL/ES complexes are functionally linked to enable substrate folding. The findings are well presented and highly significant. However, I have the following concerns to be addressed:

1. Template matching is a key method for the study. "To validate the accuracy of the template matching results", ribosomes were used to compare with the GroEL/ES complexes. Although the median ratio of

GroEL to ribosomes appears to be consistent with the results from MS, it is not clear if the template matching results are sufficiently accurate. As an example, the six particles presented in Figure 1A do not match any of the particles presented in Figure 1B.

2. The initial number of subtomograms for EL:ES1 is 176,408 while the number for the final averaging is 17,598, suggesting that 90% of the initial data were removed during the processing. It's not clear how the final subtomograms were selected.

3. Similarly, less than 10% data from EL:ES2 were selected for final averaging. It's not clear how the "bad data" were removed.

4. Given that different templates were used to find the large numbers of subtomograms for averaging, it is not clear if the structures presented in Figure 2 have sufficient resolution to avoid "reference bias" problems.

5. In Figure 3, it's impressive to see different conformations. However, the differences in Trans ring are not clearly presented at the top panels.

6. In Figure 4, the substrate appears to interact with GroEL through specific interactions. The resolution of the structures presented in panel g is too low to be sure that MetK forms the density in the GroEL:ES chamber in vivo.

Referee #3 (Remarks to the Author):

Wagner et al. use in situ cryoET to follow the compositional state and substrate occupancy of GroEL:GroES complexes in *E. coli*. In addition, single particle 3D reconstruction and in vitro cryoET are used to study substrate binding (MetK) in the GroEL chamber and compare with lower resolution states observed by in situ cryoET.

This is a beautifully executed study that allows, for the first time, to integrate the wealth of insights from in vitro GroEL studies, structural and biochemical, with in situ molecular imaging.

The technical execution and description of the structural work are state-of-the art. There is a few, mostly minor points in text and conclusions that require some clarification.

- Ln 192-92 and (Extended Data Figs. 5d and 6a-d). The authors write the 2D classes for the GroEL:GroES – MetK complexes consist of EL:ES2 and well as "a minor amount of EL:ES1 complexes". Calling a 1:2 ratio a minor amount is misleading.

- Fig. 3c gives the relative abundance of the different states of the GroEL:ES species found in in situ during 37C, HS and MetK conditions.

The figure suggests that the calculations were based on side view particles, correct? This should be mentioned.

- Ext. Data Fig 5d. The representative 2D classes for step IV, 3D alignment on the SP, show top views only. The figure legend and methods (Ln 611-613) mention manual turning of particles by $2\pi/7$ equivalents to obtain a single frame of reference.

How were side and tilted views treated? Were $2\pi/7$ states discernible here as well?

Please also provide a figure with the angular distribution of particles that went into the 3D reconstruction of MetK.

Note that the CryoDRGN panel under step IV is colored green rather than black.

- In the discussion, the authors infer or allude to the kinetics of interconversions of different GroEL:GroES states, i.e. Ln 263-264 or Ln 267-268. They likely make these inferences based on relative amounts of the complexes. However plausible, there is no direct data to allow such assertive statements. This should be toned down, or better, the authors could try build a mathematical model using interconversion rates known from in vitro studies, and see if the calculated ratios of different species match these observed in situ.

Author Rebuttals to Initial Comments:

We thank the referees for the very positive assessment of our manuscript and for their helpful comments and constructive criticism.

Referees' comments:

Referee #1 (Remarks to the Author):

In this pioneering work, Hartl and collaborators use cryo-ET to examine the folding cycle of the GroEL/S chaperonin *in vivo*. Both the type of questions asked and the interpretation of results benefit from Hartl's intimate knowledge of the *in vitro* GroEL folding literature, much of it contributed by his own laboratory. Additionally, the types of comparisons made and state of the art data analysis solidified significant conclusions--e.g. comparing groEL number to ribosome number, and undertaking specific *in vitro* experiments to allow extrapolation between data sets to establish which complex contain "folded substrate" in the GroEL complexes *in vivo*. I believe that this work is the leading edge of a new series of investigations that compare previously determined *in vitro* paradigms to actual *in vivo* function of cellular machines. This makes it very important to consider carefully how the work is presented. The comments below are suggestions when you revise the ms.

1. The basic framing of the manuscript is that there are many unsolved issues from the *in vitro* literature that this *in vivo* study will clarify. Are you sure that this is the best framing for your pioneering work, rather than documenting and rationalizing where the *in vitro* and *in vivo* results converge and differ. For example, although there have certainly been results in the literature supporting function either as a "bullet" or a "football", most of these investigations were many years ago, and at least my interpretation is that currently, most people believe that both types exist, and there is some evidence that the asymmetric football form is preferred. Note that if you agree with my impression of the current literature, I do not feel that it detracts at all from the significance of your results, but instead enhances them. For example, your finding of the distribution of complexes in exponentially growing cells is very interesting. It shows that essentially all GroEL's *in vivo* have the potential to be active, and that even though there is a 2:1 ratio of GroES to L, not all complexes are footballs, a fact explained later by the slow binding of the 2nd groEL. By focusing the ms. on differences instead of novelties, the authors may actually lose an opportunity to explain the significance of their data.

Thank you for your positive assessment. We also appreciate your suggestions for how to best present the results in light of the current state of the field. We have made changes in the Abstract and Introduction to better reflect the focus of our study on understanding how the available *in vitro* data apply to the chaperonin reaction cycle *in vivo*.

2. The author's route to identifying masses in GroEL as folded substrates is a strength of the ms (Fig. 3,4). This point could be further validated by comparing *in vivo* and *in vitro* cryoET of

MetK for a viable folding defective GroEL mutant such as the GFP-enhanced variants described in Weissman's Cell 2002 111: 1027-39 paper to see if they also track *in vivo* and *in vitro*.

We carefully considered this suggestion and thank the reviewer. However, we believe that the proposed experiments are unlikely to result in conclusive results in the context of the present study, for the following reasons:

Firstly, the GroEL/ES variant 3-1, described in the paper by Wang et al. (Cell 2002) was selected for improved folding of GFP at 37°C and allows *E. coli* cells to grow under these conditions. This would suggest that all obligate GroEL/ES substrates with essential functions must still be able to fold, which includes the GroEL substrate MetK analyzed in our study. Thus, we must assume that MetK (and many other proteins) are still folded with this GroEL mutant.

Secondly, even if the efficiency of MetK folding by GroE3-1 would be somewhat reduced, cryo-EM (both single particle analysis *in vitro* and cryo-ET *in situ*) would be unlikely to reveal structural details of unfolded MetK encapsulated in the GroEL-ES cavity. We conclude this from our present single particle analysis data set, which included a substantial number of GroEL-ES chambers containing not-yet folded MetK (note that occupancy of GroEL with MetK was demonstrated biochemically). These chambers appeared very similar to those of empty GroEL-ES (Extended Data Figs. 7b and 8b-d).

In other words, at this point the available cryo-EM methods appear to be unable to resolve unfolded or partially folded states of MetK reliably, because these intermediates do not assume a defined position within the chamber. We agree, however, that this is an important issue. Ongoing studies in the Hartl lab using mutationally destabilized GroEL substrates may succeed in detecting such states, but this is a separate project outside the scope of the present study.

Minor points.

1. L 108-110. Incorrect as stated--the authors neglect to consider that there is extensive regulation of the amount of proteins in an operon to maintain preferred ratio of subunits (Cell. 2014 Apr 24;157(3):624-35. doi: 10.1016/j.cell.2014.02.033.). However, in this case, according to the data in that ms, GroES and GroEL are produced in a 1:1 ratio, but not because it is a consequence of operon organization--rather the cell has decided that this is optimal.

We apologize for our misconception and have revised this statement (L 109-112). The reviewer is correct that presence of genes in an operon does not necessarily translate into equal abundance of gene products, although this seems to be the case for GroEL and GroES, resulting in a ~2:1 ratio of GroES 7-mer to GroEL14-mer complexes (also see the integrated *E. coli* protein abundance dataset [<https://pax-db.org/dataset/511145/2297923011>], wherein GroEL and GroES subunit abundance values are 5112 and 4556 ppm, respectively). Note that due to the small size of GroES, iBAQ values from mass spectrometry are not suitable to reliably determine its absolute abundance. We therefore used the iBAQ data only to measure the relative increase of GroEL and GroES upon heat stress or overexpression in conjunction with immunoblotting (Extended Data Fig. 3a-c).

2. L117-118. EL:ES1 complexes predominate in ALL conditions tested.

Corrected.

3. Growth experiments.

(1) Were cells in all experiments harvested at a comparable OD? This is important because cell steady state constantly changes during growth in LB, and this could reflect EL/ES state. Could the change in EL/ES after heat shock actually sampling of culture closer to stationary phase than in the other growth conditions. This should be clarified in the methods

Yes, cells were harvested for structural analysis in the exponential growth phase at approximately OD₆₀₀ 0.4 (L 419-420). For growth experiments, the cultures were diluted to OD₆₀₀ 0.1 with fresh, temperature-adjusted medium when the OD₆₀₀ just exceeded 0.4 to ensure that the cells were still in the exponential growth phase, as described in Methods (L 337-339).

(2) It would be nice to have growth data for overexpression of GroEL/S + MetK to determine whether cells are in a significantly impaired state. Also note that ribosome number changes with growth rate.

We performed the growth analysis for the strain overexpressing GroEL/ES and MetK (referred to as MetK cells) as suggested. Cell growth was measured upon expression of MetK, i.e. after expression of GroEL/ES (as for biochemical and cryo-ET analysis). The cells showed no major growth impairment with a doubling time of ~34 min (parent strain ~25 min) (new Extended Fig. 3e). Note that maintenance of high-copy number plasmids, antibiotic resistances and overexpression of proteins generally results in somewhat slower growth. We noticed that ~30% of the heavily overexpressed MetK became insoluble, i.e. the GroEL machinery (and/or other chaperones) was saturated as intended, consistent with the high occupancy of GroEL complexes with MetK detected biochemically. See also point 4 below.

4. The authors state that after GroEL overexpression, the ratio of GroEL/GroES1 to ribosomes is comparable to that in wt cells. Does that mean that there is a 40% reduction in the amount of total GroEL/S complexes?

Yes, this is correct, and thank you for pointing this out. Despite this reduction in overall GroEL:ES complexes, cell growth was almost as WT, even upon overexpression of substrate protein MetK (see below). This is now stated in the revised manuscript (L 117-121). We assume that the free GroEL may function as a buffer, consistent with our finding that free GroEL in EL+ cells has substrate bound (Fig. 2d).

It would be interesting to examine the chaperonin properties (ie =Ext. fig 4g), and cellular response MetK overproduction--substrate folding properties of this strain (ie. Figs 3,4) to see whether any of the measurable parameters of the reaction are perturbed.

We performed the growth analysis for EL+ cells with additional MetK overexpression as suggested. We measured the growth during expression of MetK, i.e. after overexpression of GroEL as used for cryo-ET analysis (also see point 3 above). The EL+ cells, with or without overexpression of MetK, grow similar to the strain overexpressing GroEL/ES and MetK (new Extended Fig. 3e). Thus, neither of these conditions

results in a major growth impairment, and, importantly, MetK overexpression in EL+ does not reduce growth significantly. The relatively slower growth of the EL+ strain in this experiment compared to Extended Data Fig. 3d (former Fig. 1c) is attributed to the fact that in Extended Data Fig. 3e the cells contain the additional construct for MetK overexpression from the T7 promoter and an additional antibiotic (ampicillin), although MetK was not induced (as we confirmed biochemically).

We attempted to classify EL:ES₁ complexes in EL+ cells with regard to the conformation of the *trans*-ring. Probably because of the low particle number (~4000 particles versus 17600 in the combined data of the other three conditions) an unbiased approach failed. Using a classification approach with either wide (left) or narrow (right) EL:ES₁ complexes as starting reference (see Figure below), we found two classes at low resolution resembling the narrow conformation, but none with a clearly wide conformation in the EL+ dataset. (Of note, a substantial proportion of particles (~ 1300) was not stably assigned.) This preliminary result is in accordance with our proposed model of the GroEL/ES reaction cycle as the relative lack of GroES in the EL+ condition would lead to an accumulation of the substrate bound, narrow *trans*-ring conformation of the EL:ES₁ complex.

Referee #2 (Remarks to the Author):

Wagner et al. present a comprehensive study that reveals distinct conformations of the bacterial chaperonin GroEL and its cofactor GroES in their native cellular environment by combining cryo-focused ion beam (cryo-EIB) milling, cryo-electron tomography (cryo-ET), and advanced subtomogram analysis. The authors show evidence that the GroEL/ES complexes co-exist in asymmetrical (EL:ES1) and symmetrical (EL:ES2) forms in vivo under different growth conditions including heat stress and high level of substrate proteins. Furthermore, the authors determine in-situ structures of the GroEL/ES complexes together with folded substrate MetK. The in-situ structures are validated by high-resolution cryo-EM structures in vitro. Overall, the manuscript not only provides the most detailed view of the GroEL/ES assisted protein folding cycle in vivo, but also argues somewhat convincingly that both asymmetrical and symmetrical forms of the GroEL/ES complexes are functionally linked to enable substrate folding. The findings are well presented and highly significant.

We thank the reviewer for his/her positive evaluation.

However, I have the following concerns to be addressed:

1. Template matching is a key method for the study. “To validate the accuracy of the template matching results”, ribosomes were used to compare with the GroEL/ES complexes. Although the median ratio of GroEL to ribosomes appears to be consistent with the results from MS, it is not clear if the template matching results are sufficiently accurate. As an example, the six particles presented in Figure 1A do not match any of the particles presented in Figure 1B.

Thank you for noting this mistake. We had accidentally used the mirror image of the tomogram for the 3D rendering in Fig. 1b. This has now been corrected and the corresponding chaperonin particles are highlighted in both panels a and b (black circles).

Regarding the general accuracy of template matching, we are aware of the limitations of the method. This is why we used orthogonal mass spectrometry data to verify our results. To alleviate potential limitations in precision as well as recall, we purposefully chose a non-stringent threshold in template matching to avoid false negative identifications and subsequently very carefully removed false positive identifications by rounds of 3D classification (see points 2 and 3).

2. The initial number of subtomograms for EL:ES1 is 176,408 while the number for the final averaging is 17,598, suggesting that 90% of the initial data were removed during the processing. It's not clear how the final subtomograms were selected.

We deliberately over-picked putative particles with low-resolution templates in the initial stage to avoid false negative assignments. Subsequently, 3D classes without emergent higher resolution features were discarded. All classifications were done repeatedly with different, randomly chosen initial starting sets of 250-500 subtomograms to generate the initial reference particles (L 495-498). Only particles that ended up in the same class for all independent rounds of classifications were retained.

To clarify the selection criteria and classification workflow we prepared the new Supplemental Figure 1 (shown below for the convenience of the reviewer).

Supplementary Figure 1: Classification of GroEL:ES particles after template matching. (a) Templates. Central slices of the 3D volume of the templates (PDB entry 1AON was used for EL:ES₁, 4PKO for EL:ES₂) used in the final round of template matching in side and top view. (b) Class selection criteria. A top view of the central slice of the EL:ES₁ template at higher magnification is shown on the left. A class that was selected for further processing is shown in the middle and, on the right, one that was discarded as false positive. The white arrow highlights the feature chosen

as selection criterion: Detail in the structure in the equatorial domain of the GroEL that is not visible at the resolution of template matching, but is clearly detected in the classes containing true positive particles. (c) Classification workflow. Detailed overview of the class selection process for the processing workflow described in Methods and summarized in Extended Data Fig. 1b. Classes selected for further processing are highlighted by a green outline, while classes that were discarded as false positives are marked with a red outline. As described in Methods, all classifications were performed multiple times and only particles that ended up in the same class in all independent runs were considered for further processing, as previously described in (Erdmann, P. S. et al. In situ cryo-electron tomography reveals gradient organization of ribosome biogenesis in intact nucleoli. *Nat Commun* 12, 5364 (2021).)

3. Similarly, less than 10% data from EL:ES2 were selected for final averaging. It's not clear how the "bad data" were removed.

Please see response to point 2.

4. Given that different templates were used to find the large numbers of subtomograms for averaging, it is not clear if the structures presented in Figure 2 have sufficient resolution to avoid "reference bias" problems.

For every template matching conducted, we cross-identified the other species: Regardless of whether we used EL:ES₁ or EL:ES₂ as the search template, we were able to identify both a class resembling EL:ES₁ particles and a class resembling EL:ES₂ particles (Supplemental Fig. 1c, green classes in the middle row). This indicates that reference bias was not a major problem. The refined resolution values of the reconstructions (~12 Å) were well below the resolution of the search templates (40 Å). Moreover, the wide *trans* rings in EL:ES₁ complexes, apparent in the ~12 Å reconstruction, were not present in the search templates.

For the EL+ dataset, we performed the first round of template matching with the reference structures derived from our other datasets. These only contained EL:ES₁ and EL:ES₂ and were filtered to a resolution of 40 Å. In the first rounds of classification, we then identified the EL tetradecamer species depicted in Fig. 2d and then performed an additional round of template matching with a data-derived reference. As a result, although no apo-GroEL reference was used, we were able to refine a corresponding structure to a resolution of ~10 Å.

Furthermore, we calculated the resolution-dependent cross correlation between our final molecular model and the final map as well as between our final molecular model and the initial search template. The Figure below shows that the model only modestly correlates with the search template (33.5 Å), but correlates well with the final structure up to a resolution of 13.4 Å. Therefore, our template matching classification and averaging workflow led to a gain of around ~20 Å in resolution that cannot be explained by model bias.

5. In Figure 3, it's impressive to see different conformations. However, the differences in Trans ring are not clearly presented at the top panels.

In the side views of the *trans*-rings, the major difference between conformations is the presence of a central substrate density. A white arrowhead points to this density to clarify the differences between SP-bound and non-SP-bound classes (Figure 3a).

6. In Figure 4, the substrate appears to interact with GroEL through specific interactions. The resolution of the structures presented in panel g is too low to be sure that MetK forms the density in the GroEL:ES chamber in vivo.

Figure 4g represents a mixture between MetK and other chaperonin substrates, although strongly dominated by the highly overexpressed MetK (confirmed by biochemical analysis; Extended Data Fig. 3e, f). This is now clarified (L 239-243). The different appearance compared to panels a-d is a consequence of the applied seven-fold rotational averaging. The particle numbers and the resolution were too low to resolve the asymmetry of the substrate interactions with the chaperonin cage in cryo-ET. This is now better explained in the text (L 235, 958-959).

Of note, we independently solved at high resolution the structures of GroEL:ES with two other folded substrate proteins encapsulated in the chaperonin cage (unpublished data). Both exhibit similar locations in the lower parts of the cage. Thus, we are confident that the density inside the chamber in Fig. 4g represents structurally ordered substrate protein.

Referee #3 (Remarks to the Author):

Wagner et al. use in situ cryoET to follow the compositional state and substrate occupancy of GroEL:GroES complexes in *E. coli*. In addition, single particle 3D reconstruction and in vitro cryoET are used to study substrate binding (MetK) in the GroEL chamber and compare with lower resolution states observed by in situ cryoET.

This is a beautifully executed study that allows, for the first time, to integrate the wealth of insights from in vitro GroEL studies, structural and biochemical, with in situ molecular imaging.

The technical execution and description of the structural work are state-of-the art. There is a few, mostly minor points in text and conclusions that require some clarification.

Thank you for the positive evaluation.

- Ln 192-92 and (Extended Data Figs. 5d and 6a-d). The authors write the 2D classes for the GroEL:GroES – MetK complexes consist of EL:ES2 and well as “a minor amount of EL:ES1 complexes”. Calling a 1:2 ratio a minor amount is misleading.

We agree and have changed the wording accordingly (L 197-198).

- Fig. 3c gives the relative abundance of the different states of the GroEL:ES species found in in situ during 37C, HS and MetK conditions.

The figure suggests that the calculations were based on side view particles, correct? This should be mentioned.

Fig. 3a-c shows central slices through subtomogram averages, not 2D classes of particles. We chose this representation to more clearly display the substrate density inside the chaperonin chamber and at the opening of the *trans* ring. The calculations are thus based on all particles in the respective subtomogram class.

To highlight that subtomogram averages are shown, we now provide this information in the Figure legend (L 923-924, 928-930, 934-936).

- Ext. Data Fig 5d. The representative 2D classes for step IV, 3D alignment on the SP, show top views only. The figure legend and methods (Ln 611-613) mention manual turning of particles by $2\pi/7$ equivalents to obtain a single frame of reference.

How were side and tilted views treated? Were $2\pi/7$ states discernible here as well?

We apologize for the lack of clarity. These are not 2D classes but a representation of 3D volumes as central horizontal slices through the density. The 3D classification step was carried out starting from aligned chaperonin chambers. The subsequent manual $n \cdot 360^\circ/7$ rotations were applied to the 3D class averages and additionally to the particle orientations within each class. We added “This was done by adding the corresponding increment to the particle rotation angles in the particle table (.star file).” to lines 627-628 in Methods to clarify this point.

Please also provide a figure with the angular distribution of particles that went into the 3D reconstruction of MetK.

A representation of the angular distribution for the fully aligned final MetK was added as an additional figure panel (new Extended Data Fig. 7g). There were no strongly preferred orientations of MetK in the sample.

Note that the CryoDRGN panel under step IV is colored green rather than black.

Thank you for noting this. Corrected.

- In the discussion, the authors infer or allude to the kinetics of interconversions of different GroEL:GroES states, i.e. Ln 263-264 or Ln 267-268. They likely make these inferences based on relative amounts of the complexes. However plausible, there is no direct data to allow such assertive statements. This should be toned down, or better, the authors could try build a mathematical model using interconversion rates known from *in vitro* studies, and see if the calculated ratios of different species match these observed *in situ*.

We initially hoped to create a Markov chain model for the reaction system and then write a differential equation for every possible transition. However, our system is underdetermined (nucleotide states, local concentrations, ATP/ADP ratio are unknown) and we observe additional conformational states for the EL:ES₁ complex – with narrow and wide *trans* rings – that were not considered in prior *in vitro* studies.

We replaced “rate-limiting” with “limiting” in “Our data suggest that SP binding to the EL:ES₁ trans-ring, stabilizing a narrow ring conformation, is limiting for EL:ES₂ formation.” (L 269-270)

We replaced “However, binding of SP in *trans* and subsequent association of the second GroES is slow.” with “However, binding of SP in *trans* and subsequent association of the second GroES appears to be relatively slow.” (L 273-274)

Reviewer Reports on the First Revision:

Referees' comments:

Referee #1 (Remarks to the Author):

Thank you for the additional clarifying experiments that you performed, which satisfied both my major and minor comments.

Upon rereading the ms. I would like clarification on one point. You note in the text that following HS, the ratio of

EL:ES 1 complexes increased to 70% (Fig 1c). However, it appears that the EL:ES ratio also increases slightly in HS conditions (extended fig 3C). If correct, might the change in ratio account for increased prevalence of EL/ES1?

Referee #2 (Remarks to the Author):

The authors provided additional information in the revised manuscript to address the concerns from three reviewers. However, the new supplementary figure 1 did not address the previous concern. In contrast, the figure raise more concerns about the potential reference bias. Specifically, the round 1 classification removes ~75% of the particles which were initially selected by template matching. Although it seems reasonable to discard 3D classes without higher resolution features, none of them looks different from the template. In addition, the discarded classes have features related to missing wedge artifacts which are normal for tomography data.

Given that the authors "deliberately over-picked putative particles", I am surprised by the supplementary figure 1 in which every class is similar to the template. If the real particle number for EL:ES1 is about 17,598, it's not clear why 90% of the selected particles also look like the template.

Referee #3 (Remarks to the Author):

The authors clarify most of points raised by the referees.

I have a few remaining points with the model presented by the authors in Fig. 5. Whilst the cryoET analyses provide an unprecedented insight in the relative ratios of the different GroEL:ES particles present during normal physiology and unfolded protein stress conditions, it does not in itself inform on a folding trajectories or relative rates of the different steps. This is only possible by reference to in vitro studies, as is done in the discussion. Some points of the model and discussion still need some clarification or nuance:

In L 255-256 the authors write “SP exits the folding chamber upon GroES dissociation, facilitated by a wide conformation of the apical domains, possibly generating a short-lived GroEL-only intermediate”. Since there is no evidence in present study that a wide trans domain facilitates GroES dissociation, add the references) to the negative allostery (1, 44, 45) or leave out this part of the sentence. This point is made Lns 261-266.

In (L 269-270) “Our data suggest that SP binding to the EL:ES1 trans-ring, stabilizing a narrow ring conformation, is limiting for EL:ES2 formation.”

Are the authors implying any priority in SP binding versus the wide to narrow conversion of the EL:ES1 trans ring? The reader may interpret this as such.

In their heath shock experiments, the authors observe an increase in EL:ES1 trans wide over EL:ES1 narrow or EL:ES2, which they argue to be based on altered ATP/ADP ratios. This seems plausible. This observation, and the EL:ES particle ratios observed by the authors at 37C and particularly during MetK overexpression would argue that the limiting step in EL:ES1-SP to EL:ES1– SP2 and on to EL:ES2–SP2 particles is not substrate binding, but the wide to narrow conversion of the trans ring. The heath shock and MetK experiments show that excess folding substrate does not in a significant way alter the conversion rate. It rather appears that the ADP/ATP ratios, the negative allostery with the Cis chamber, and possibly other cellular factors limit the EL:ES1 to EL:ES2 progression. Also the more hydrophobic nature of the narrow conformation would argue for this to be the form that recruits substrate, and thus SP binding being secondary to wide to narrow conversion of the trans ring.

Minor point:

In extended data table 1a and 1b, please add the model-map CC values (CCvolume, CCmask, CCpeaks) from Phenix.refine.

Comments of reviewer #2 on the provisional rebuttal letter to their outstanding concerns:

"...I also very much appreciate the responses from the authors. It's clear that the authors are experts in the field. I have no concern on their expertise and excellence. Unfortunately, they did not address my concerns on their supplementary figure 1 and their approach. As I said before, the discarded classes look "normal" although they have features resulted from missing wedge artifacts. In addition, their resolution is not significantly different from the good class (green). Therefore, the resolution is not a reliable parameter for selecting "good classes". The authors are correct. It is reasonable to "set a less stringent threshold during template matching" and "fully anticipating the presence of a significant number of false positives, or noise volumes". However, it's more critical to remove "junk" by using sophisticated classification procedures or more reliable criteria. None of discarded classes looks "bad" to me.

More importantly, it is common to use one reference for template matching in order to identify new features that are not present in the original reference. As an example, if the authors use EL:ES1 as the only reference, they should be able to find EL:ES1 and EL:ES2 particles. The EL:ES2 particles should be reliable as they have extra densities that are not present in EL:ES1. Similarly, if the authors use EL:ES2 as the only symmetric reference, they should be able to find EL:ES1 particles which are asymmetric.

Given that the reference bias is likely present in the current template matching approach, more cautious analyses and validation should be considered for such an important subject."

Author Rebuttals to First Revision:

Response to reviewers on revised manuscript

Referee #1 (Remarks to the Author):

Thank you for the additional clarifying experiments that you performed, which satisfied both my major and minor comments.

Upon rereading the ms. I would like clarification on one point. You note in the text that following HS, the ratio of EL:ES 1 complexes increased to 70% (Fig 1c). However, it appears that the EL:ES ratio also increases slightly in HS conditions (extended fig 3C). If correct, might the change in ratio account for increased prevalence of EL/ES1?

Thank you for raising this interesting point. Indeed, we detected a slight increase in the ratio of GroEL:GroES upon heat stress (HS) and MetK overexpression by mass spectrometry (37 °C: 0.57 ± 0.11 mean \pm std; HS: 0.69 ± 0.12 mean \pm std, MetK: 0.83 ± 0.09 mean \pm std). The differences between the groups were however not statistically significant when compared with a 1-way ANOVA test (although the difference between 37 °C and MetK is at the border of significance). This is now stated in the figure legend.

We think that it is unlikely that the observed increase of the EL:ES₁ fraction under HS is caused by changes in subunit expression levels, as we would expect a similar increase in the MetK condition, but actually find a decrease (Fig. 1c). As discussed, we consider it more likely that a change in ADP/ATP concentration ratio contributes to the higher population of EL:ES₁ complexes under HS.

Referee #2 (Remarks to the Author):

The authors provided additional information in the revised manuscript to address the concerns from three reviewers. However, the new supplementary figure 1 did not address the previous concern. In contrast, the figure raise more concerns about the potential reference bias. Specifically, the round 1 classification removes ~75% of the particles which were initially selected by template matching. Although it seems reasonable to discard 3D classes without higher resolution features, none of them looks different from the template. In addition, the discarded classes have features related to missing wedge artifacts which are normal for tomography data.

Given that the authors "deliberately over-picked putative particles", I am surprised by the supplementary figure 1 in which every class is similar to the template. If the real particle number for EL:ES1 is about 17,598, it's not clear why 90% of the selected particles also look like the template.

First response to reviewer

The issue raised by the reviewer likely stems from the "Einstein from noise" problem¹⁻³, which we have been aware of since introducing template matching. This occurs when a large number of noisy images are correlated with a template, and upon averaging will eventually reproduce the template. A rigorous safeguard against this is to filter the reference at low resolution and perform "gold standard" processing where the data is split into two

independent half sets, as we did for this work^{4,5}. With this approach, finding consistent structural features beyond the low resolution of the template should only be possible from actual particles and not from noise.

Our decision to set a less stringent threshold during template matching was strategic, fully anticipating the presence of a significant number of false positives, or noise volumes, within our dataset. Such volumes may encompass genuine false positives not corresponding to EL:ES particles, as well as incomplete or damaged EL:ES complexes at the lamella surface. These volumes would likely resemble the template up to the resolution limit imposed by the low-pass filter when averaged due to the “Einstein from noise” effect. We chose this trade-off to limit the possibility of a significant fraction of false negative GroEL particles in our dataset, as this could not be remedied in further steps. False positives on the other hand can be eliminated later in the processing pipeline by classification, potentially after a more exhaustive angular search and at a binning that allows for the emergence of higher resolution features.

This approach is common practice in subtomogram averaging⁶⁻¹¹, due to computational reasons. Template matching is extremely computationally expensive if a fine angular sampling at a higher resolution is chosen to find particles. Even with the relatively coarse-grained angular sampling of 15° at four times binning, as used in this work, template matching of one tomogram took around 1 day on 120 CPU cores. Classification on the other hand has a much smaller search space due to the reduced volume (64x64x64 pixel box size for a GroEL vs 1800x1800x512 pixel box size for a tomogram, both in bin2), allowing for finer angular sampling during alignment and classification.

Given the substantial computational cost of subtomogram averaging, we elected to only refine our particle orientations using a local angular search. This is reasonable as template matching is already a rough alignment approach, so only local angular searches are required to determine the true orientations. While this is sufficient to align higher resolution structural features in true positive classes, it is also insufficient to cause the noise classes to diverge from the template structure, as false positives are likely trapped in local minima.

To further substantiate this point, we conducted a preliminary experiment where the angles of EL:ES₂ particles were randomized, comparing the final set of particles with the set of particles discarded in the first round of classification (see Reviewer Fig. 1). After a global alignment, the final particle set converged to an EL:ES₂ structure, while the discarded set of particles did not converge to any recognisable structure. These data strongly suggest that our selection of true and false positives is justified.

Reviewer Fig. 1: Averages of EL:ES₂ particle sets after angular randomization and global alignment. The final, “true positive” set of EL:ES₂ particles (**left**; gray) and all discarded particles from the first round of classification (**right**; yellow) were assigned random initial Euler angles between -180° and 180°. The sets were averaged and used as an initial reference for a global alignment in RELION 3.0. While true positive EL:ES₂ particles converged to a recognizable EL:ES₂ structure, the discarded particles did not converge into any recognizable shape. Both structures were rendered in ChimeraX at a contour level of 0.13. Differences from the structures presented in the main Figures are due to the fact that here the alignment was done without imposed symmetry, masking or initial reference.

Altogether, while we cannot prove that none of the discarded particles is a true positive, strong evidence indicates that the particles selected are true positives and contribute structural information far beyond the resolution of the template:

- An EL:ES₁ class was found using the EL:ES₂ template and vice versa, while the discarded classes only showed similarity to a low-resolution version of the template (Supplemental Figure S1).
- EL complexes lacking ES were found using an EL:ES₁ template (EL+ condition).
- We achieved final resolutions of 10-12 Å, starting with templates of 40 Å resolution.

Lastly, we want to further emphasize that discarding a significant fraction of the detected particles is standard practice, even for complexes far more readily detectable like the ribosome. For example, Beck et al 2023⁷ described the following:

“In total, 286,400 untreated ribosome sub-tomograms and 281,600 HHT-treated were reconstructed.” “Sub-tomograms from template matching were classified and refined in RELION 3.1 (61). 39,402 particles from the untreated dataset and 39,070 from the treated dataset were assigned to the 80S ribosome class.”

As a second example, in this work⁸ Beck et al. used only under half of the detected ribosomes for averaging and classification and discarded the rest of the picks.

Comments of reviewer #2 on the provisional rebuttal letter to their outstanding

concerns:

"...I also very much appreciate the responses from the authors. It's clear that the authors are experts in the field. I have no concern on their expertise and excellence. Unfortunately, they did not address my concerns on their supplementary figure 1 and their approach. As I said before, the discarded classes look "normal" although they have features resulted from missing wedge artifacts. In addition, their resolution is not significantly different from the good class (green). Therefore, the resolution is not a reliable parameter for selecting "good classes". The authors are correct. It is reasonable to "set a less stringent threshold during template matching" and "fully anticipating the presence of a significant number of false positives, or noise volumes". However, it's more critical to remove "junk" by using sophisticated classification procedures or more reliable criteria. None of discarded classes looks "bad" to me.

We thank the reviewer for his/her helpful feedback. In the revised manuscript we include a modified version of Supplemental Fig. 1. We acknowledge that we could have explained the particle selection criteria more clearly, given that the discarded classes resemble the EL:ES structure due to the "Einstein from noise" effect, as explained in the previous response. To clarify the reasoning behind the feature utilized for selection we introduce the new Supplemental Fig. 1a:

In previous crystal and cryoEM structures of GroEL/ES and bacterial chaperonins, the equatorial domain rings are always the best-defined (i.e. most rigid, as judged from atomic B-factors) and most uniform regions, while the intermediate and apical domains can undergo extensive reorientations. Thus, the equatorial domain rings should be the best-defined part in 3D classes of true positive particles, and the bilobal organization of the equatorial domains should become apparent in the cross-section as the first higher resolution feature to serve as selection criteria (new Supplemental Fig. 1a, see arrowheads on the crystal structures of EL:ES₂ [PDB 4PKO] and EL:ES₁ [PDB 1AON] filtered to 14 Å). While this feature is visible in an exemplary selected 3D class, it is not discernible in the template and in an exemplary discarded class (new Supplemental Fig. 1a, arrowheads).

Therefore, while the discarded classes may look "normal", as noted by the reviewer, they do not display the emergent higher resolution feature which was our selection criterion. Selecting only classes showing this emergent higher resolution feature is an objective and reproducible method of class selection, avoiding false positives.

We also would like to emphasize once more that we independently validated the quantification of chaperonin complexes and ribosomes from cryo-ET using biochemical analysis and mass spectrometry. Both data sets are in good agreement (Extended Data Fig. 2c), ruling out the possibility of a major error in the quantification of chaperonin particles by cryo-ET.

Furthermore, we would like to refer to the validation experiment in our initial response to the comments of referee #2 (Reviewer Fig. 1 above). This analysis provides strong supporting evidence that there is a clear distinction between the discarded classes and the selected particles. After randomization of the initial angles, removing prior information, only the particles belonging to the final EL:ES₂ class recovered clear structural features, whereas the particles belonging to the discarded classes did not align to any discernible structure (see Reviewer Fig. 1 above).

More importantly, it is common to use one reference for template matching in order to identify new features that are not present in the original reference. As an example, if the authors use EL:ES1 as the only reference, they should be able to find EL:ES1 and EL:ES2 particles. The EL:ES2 particles should be reliable as they have extra densities that are not present in EL:ES1. Similarly, if the authors use EL:ES2 as the only symmetric reference, they should be able to find EL:ES1 particles which are asymmetric.

We agree with Reviewer #2 that this is an important experiment, which we included already in the original version of the manuscript and highlighted in our response. Indeed, as indicated, we were able to detect EL:ES₁ particles with the EL:ES₂ template and vice versa, thereby addressing the concerns of the Reviewer.

At the same time, it is also true that our classification strategy did not identify all EL:ES₁ and EL:ES₂ particles independent of the reference template used. However, as pointed out in our previous response, we did cross-identify in classification round 2 a significant fraction of the other particle class in both the EL:ES₁ (~57%) as well as EL:ES₂ branch (~60%). The lack of perfect cross identification is most likely due to our very stringent exclusion criteria, where only particles being consistently assigned to the same class in all 5 repeats of the classification were accepted, as previously described in ¹². So even small differences in averages in both template matching branches will lead to differences in assignment of the noisy subtomograms.

Given that the reference bias is likely present in the current template matching approach, more cautious analyses and validation should be considered for such an important subject."

We agree that a certain bias in the template matching approach may be inevitable, and we now mention this explicitly in the main text (lines 84-86). However, such bias would have equally affected the different particle classes analyzed and would not have distorted their quantitative relation. We conclude from the agreement of cryo-ET and biochemical analysis that any existing bias must be rather limited.

We hope that our response and additional analyses address the concerns raised by reviewer.

Referee #3 (Remarks to the Author):

The authors clarify most of points raised by the referees.

We thank the referee for his/her careful reading and helpful suggestions.

I have a few remaining points with the model presented by the authors in Fig. 5. Whilst the cryoET analyses provide an unprecedented insight in the relative ratios of the different GroEL:ES particles present during normal physiology and unfolded protein stress conditions, it does not in itself inform on a folding trajectories or relative rates of the different steps. This is only possible by reference to in vitro studies, as is done in the discussion. Some points of the model and discussion still need some clarification or nuance:

In L 255-256 the authors write "SP exits the folding chamber upon GroES dissociation, facilitated by a wide conformation of the apical domains, possibly generating a short-lived GroEL-only intermediate".

Since there is no evidence in present study that a wide trans domain facilitates GroES dissociation, add the references) to the negative allostery (1, 44, 45) or leave out this part of the sentence. This point is made Lns 261-266.

We apologize for the lack of clarity. We rephrased the sentence in L 238-240 to avoid ambiguity: "After GroES dissociation, SP exits the folding chamber, facilitated by a wide conformation of the GroEL apical domains." In this phrasing, the wide conformation unambiguously facilitates SP exit, not GroES dissociation.

In (L 269-270) "Our data suggest that SP binding to the EL:ES1 trans-ring, stabilizing a narrow ring conformation, is limiting for EL:ES2 formation."

Are the authors implying any priority in SP binding versus the wide to narrow conversion of the EL:ES1 trans ring? The reader may interpret this as such.

In their heath shock experiments, the authors observe an increase in EL:ES1 trans wide over EL:ES1 narrow or EL:ES2, which they argue to be based on altered ATP/ADP ratios. This seems plausible. This observation, and the EL:ES particle ratios observed by the authors at 37C and particularly during MetK overexpression would argue that the limiting step in EL:ES1-SP to EL:ES1-SP2 and on to EL:ES2-SP2 particles is not substrate binding, but the wide to narrow conversion of the trans ring. The heath shock and MetK experiments show that excess folding substrate does not in a significant way alter the conversion rate. It rather appears that the ADP/ATP ratios, the negative allostery with the Cis chamber, and possibly other cellular factors limit the EL:ES1 to EL:ES2 progression. Also, the more hydrophobic nature of the narrow conformation would argue for this to be the form that recruits substrate, and thus SP binding being secondary to wide to narrow conversion of the trans ring.

The reviewer is correct that we cannot determine any priority of events regarding conformational changes in the *trans*-ring and SP binding. It is indeed plausible that the wide to narrow conversion of the *trans*-ring is limiting for EL:ES₁-SP to EL:ES₁-SP2 to EL:ES₂-SP2 particle formation, governed by ADP/ATP ratios and the negative allostery with the *cis* ring. We have clarified this in the text in lines 145-146, 149-154 and 249-252.

Minor point:

In extended data table 1a and 1b, please add the model-map CC values (CCvolume, CCmask, CCpeaks) from Phenix.refine.

We joined Extended Data Tables 1a and 1b and added the requested information to the new Extended Data Table 1.

- 1 Bohm, J. *et al.* Toward detecting and identifying macromolecules in a cellular context: template matching applied to electron tomograms. *Proc Natl Acad Sci U S A* **97**, 14245-14250 (2000). <https://doi.org:10.1073/pnas.230282097>
- 2 Frangakis, A. S. *et al.* Identification of macromolecular complexes in cryoelectron tomograms of phantom cells. *Proc Natl Acad Sci U S A* **99**, 14153-14158 (2002). <https://doi.org:10.1073/pnas.172520299>
- 3 Henderson, R. Avoiding the pitfalls of single particle cryo-electron microscopy: Einstein from noise. *Proc Natl Acad Sci U S A* **110**, 18037-18041 (2013). <https://doi.org:10.1073/pnas.1314449110>

- 4 Rosenthal, P. B. & Henderson, R. Optimal determination of particle orientation, absolute
hand, and contrast loss in single-particle electron cryomicroscopy. *J Mol Biol* **333**, 721-745
(2003). <https://doi.org:10.1016/j.jmb.2003.07.013>
- 5 Wan, W., Khavnekar, S. & Wagner, J. STOPGAP: an open-source package for template
matching, subtomogram alignment and classification. *Acta Crystallogr D Struct Biol* **80**, 336-
349 (2024). <https://doi.org:10.1107/S205979832400295X>
- 6 Hoffmann, P. C. *et al.* Structures of the eukaryotic ribosome and its translational states in situ.
Nat Commun **13**, 7435 (2022). <https://doi.org:10.1038/s41467-022-34997-w>
- 7 Xing, H. P. *et al.* Translation dynamics in human cells visualized at high resolution reveal
cancer drug action. *Science* **381**, 70-75 (2023). <https://doi.org:10.1126/science.adh1411>
- 8 Khusainov, I. *et al.* The killing of human gut commensal E. coli ED1a by tetracycline is
associated with severe ribosome dysfunction. *bioRxiv*, 2023.2007. 2006.546847 (2023).
- 9 O'Reilly, F. J. *et al.* In-cell architecture of an actively transcribing-translating expressome.
Science **369**, 554-557 (2020). <https://doi.org:10.1126/science.abb3758>
- 10 Xue, L. *et al.* Visualizing translation dynamics at atomic detail inside a bacterial cell. *Nature*
610, 205-211 (2022).
- 11 Xue, L., Spahn, C. M., Schacherl, M. & Mahamid, J. Structural insights into context-dependent
inhibitory mechanisms of chloramphenicol in cells. *BioRxiv*, 2023.2006.2007.544107 (2023).
- 12 Erdmann, P. S. *et al.* In situ cryo-electron tomography reveals gradient organization of
ribosome biogenesis in intact nucleoli. *Nat Commun* **12**, 5364 (2021).
<https://doi.org:10.1038/s41467-021-25413-w>

Reviewer Reports on the Second Revision:

Referees' comments:

Referee #2 (Remarks to the Author):

I appreciate the authors' efforts to address my previous concerns. The revised manuscript looks great. This is an outstanding paper which will likely have profound impacts on cryo-EM/ET applications and chaperonin biology.

Referee #3 (Remarks to the Author):

The authors' rephrasing of text and discussion brings a more nuanced and prudent description of the cascade of events in their proposed models, which cannot unambiguously be discerned from the data.